# Generating viable mice with heritable embryonically lethal mutations using the CRISPR-Cas9 system in two-cell embryos

Yi Wu[1,2,3,8], Jing Zhang[2,8], Boya Peng[3], Dan Tian[4], Dong Zhang (ID) [4], Yang Li[2], Xiaoyu Feng[2], Jinghao Liu[5], Jun Li[5], Teng Zhang[3], Xiaoyong Liu[6], Jing Lu[1,3], Baian Chen (ID) [1,2,3] & Songlin Wang[2,7]

A substantial number of mouse genes, about 25%, are embryonically lethal when knocked out. Using current genetic tools, such as the CRISPR-Cas9 system, it is difficult—or even impossible—to produce viable mice with heritable embryonically lethal mutations. Here, we establish a one-step method for microinjection of CRISPR reagents into one blastomere of two-cell embryos to generate viable chimeric founder mice with a heritable embryonically lethal mutation, of either *Virma* or *Dpm1*. By examining founder mice, we identify a phenotype and role of *Virma* in regulating kidney metabolism in adult mice. Additionally, we generate knockout mice with a heritable postnatally lethal mutation, of either *Slc17a5* or *Ctla-4*, and study its function in vivo. This one-step method provides a convenient system that rapidly generates knockout mice possessing lethal phenotypes. This allows relatively easy in vivo study of the associated genes' functions.

[1] Department of Neurobiology, Beijing Key Laboratory of Neural Regeneration and Repair, School of Basic Medical Sciences, Capital Medical University, Beijing 100069, China. [2] Molecular Laboratory for Gene Therapy and Tooth Regeneration, Beijing Key Laboratory of Tooth Regeneration and Function Reconstruction, School of Stomatology, Capital Medical University, Beijing 100050, China. [3] Laboratory Animal Center, Capital Medical University, Beijing 100069, China. [4] Experimental and Translational Research Center, Beijing Friendship Hospital, Capital Medical University, Beijing 100050, China. [5] Laboratory Animal Center, Peking University, Beijing 100871, China. [6] Department of Oral Pathology, Beijing Stomatology Hospital, Capital Medical University, Beijing 100050, China. [7] Department of Biochemistry and Molecular Biology, School of Basic Medical Sciences, Capital Medical University, Beijing 100069, China. [8] These authors contributed equally: Yi Wu, Jing Zhang. Correspondence and requests for materials should be addressed to B.C. (email: baianchen@ccmu.edu.cn) or to S.W. (email: slwang@ccmu.edu.cn)

Gene function in vivo is routinely determined using standard knockout (KO) methods, which involve deleting the genomic sequence that encodes the gene of interest[1]. Knockouts that produce embryonic lethality signal the essential role of that lethal gene in embryonic development. In mice, about 25% of gene knockouts are embryonically lethal[2,3], but until now, there was no simple, rapid method for generating knockout mice with embryonically lethal phenotypes.

The CRISPR-Cas9 gene editing system has been used to generate knockout animals of various species through the process of directly microinjecting Cas9 mRNA and single-guide RNAs (sgRNAs) into pronuclear-stage zygotes[4]. This method outperforms all other means of genetic modification, and dramatically shortens the time required to develop genetically engineered animal models. However, for a substantial number of genes that cause embryonic lethality when knocked out, this approach fails to produce knockout founder (F0) mice. CRISPR/Cas9-mediated genome editing is so efficient that F0 mice with lethal mutations produced by this method do not survive embryogenesis. The resulting lack of F0 mice makes it impossible to transmit lethal mutations to an F1 generation and thus to generate conventional knockout animal models for studying these genes' functions in vivo.

Recently, a two-step microinjection method of generating chimeric two-cell stage embryos was established using CRISPR/Cas9-mediated mutagenesis complements. This method was used to generate fluorescent-reporter chimeric mutant mice, thus enabling the discovery of the functions of the ten–eleven translocation 3 (Tet3) gene (postnatally lethal) in adult animals[5]. However, this method tested only one postnatally lethal gene, which accounts for less than 5% of gene knockouts in mice[2,3]. In addition, this microinjection procedure is complicated, and has not yet succeeded in producing knockout mice with embryonically lethal mutations.

Thus, there is an urgent need for a method that can prolong the survival of knockout F0 mice with embryonically lethal mutations. Here, we present a one-step method for microinjecting either the mRNA or the protein of Cas9 and sgRNA into one blastomere of a two-cell embryo cultured from zygotes fertilized in vivo or in vitro. Using the one-step two-cell embryo microinjection (OSTCM) method, we generate viable adult chimeric mice with an embryonically lethal mutation of either Virma or Dpm1. These lethal mutations are successfully transmitted to F1 generations by crossing. After heterozygous crossing, we use the homozygous offspring to study gene function in vivo before the animals' death. In addition, we use the OSTCM method to generate conventional knockout mice to study, in vivo, functions of the postnatally lethal mutations of Slc17a5 or Ctla-4. This method makes it possible to generate viable chimeric F0 mice that can be used to investigate in vivo functions of genes that are otherwise embryonically or postnatally lethal when conventionally knocked out.

## Results

### F0 mice with heritable embryonically lethal mutations.
Embryonic lethality is common in mice that are homozygous for genetically engineered mutations[2]. To produce adult F0 mice carrying embryonically lethal mutations, it is important to efficiently produce monoallelic mutants using CRISPR-Cas9 technology. Previous studies have shown that microinjection of low concentrations of Cas9 mRNA/sgRNA has substantial effects on monoallelic mutation-generation efficiencies[4]. Therefore, we first chose the ten–eleven translocation 2 (Tet2) gene, targeting this locus with sgRNA as previously described[4], to screen different ratios of Cas9 mRNA/sgRNA concentrations (12.5:6.3 ng/µl,

25:12.5 ng/µl, 50:25 ng/µl, and 100:50 ng/µl) for the production of monoallelic mutants in developed blastocysts.

Restriction fragment length polymorphism (RFLP) analysis and Sanger sequencing of blastocysts showed that only one of 36 (~2.8%) of the embryos from the lowest-concentration group (12.5:6.3 ng/µl Cas9 mRNA/sgRNA) was a monoallelic mutant (Supplementary Fig. 1a–c). Given the relatively equal efficiency of nonhomologous end joining, the lowest-concentration group also demonstrated a higher rate of blastocyst development (Supplementary Table 1). Overall, microinjection of low concentrations of Cas9 mRNA/sgRNA did not efficiently and consistently generate monoallelic mutants because it was difficult to precisely microinject equal amounts of Cas9 mRNA/sgRNA into each zygote.

In an attempt to generate monoallelic mutants, we also microinjected Cas9 mRNA, Tet2 sgRNA, and a donor oligonucleotide (oligo) of the wild-type (wt) DNA sequence (Supplementary Fig. 1a), while DNA double-stranded breaks generated by CRISPR-Cas9 were being repaired by the homology-directed repair pathway[6]. Unfortunately, among the embryos microinjected with Cas9 mRNA (12.5 ng/µl), sgRNA (6.3 ng/µl), and the wild-type donor oligo (50 ng/µl), no monoallelic mutants were detected using TA cloning and Sanger sequencing analysis (Supplementary Fig. 1c), although RFLP showed some blastocysts with possible wt alleles (Supplementary Fig. 1b). These results suggested that co-microinjection of wt donor oligo DNA with the CRISPR-Cas9 system is an ineffective method to generate monoallelic mutants.

sgRNA targeting efficiency varies significantly between loci and even between target sites within the same locus[7]. We chose Virma, a gene that is embryonically lethal when knocked out[2], as the target to produce adult F0 mice carrying frameshift mutations using zygote microinjection with different sgRNA sequence features (sgRNA1, sgRNA2, and sgRNA3; Fig. 1a). Using the CRISPRko[8] prediction tool, we found that the sgRNAs' on-target efficiency scores differed (sgRNA1, 0.771; sgRNA2, 0.645; and sgRNA3, 0.450). The VIRMA protein, a key component of the mRNA N6-methyladenosine (m6A) methylosome, is involved in mRNA splicing and processing[9,10]. Although we transplanted more than 98 early two-pronuclear zygotes microinjected with Cas9 mRNA and one of the three sgRNAs into the oviducts of pseudopregnant females (>4 females per sgRNA), no mice were born, regardless of which sgRNA was used (Table 1). Also, following pronuclear fusion, 42 zygotes were microinjected with Cas9 mRNA and sgRNA1 then transplanted into the oviducts of two pseudopregnant females, yet no mice were born. Our results indicated that zygote microinjection methods using either different sgRNA sequence features or zygote development stages could not generate viable F0 mice with embryonically lethal Virma mutations. From this, we reasonably expect this method to likely be unable to produce viable F0 mice with other embryonically lethal mutations.

We reasoned that the embryonic stage at which we administered the microinjection could be changed from the zygote to the two-cell embryo. Therefore, we microinjected one blastomere of two-cell embryos to generate chimeric mice with embryonically lethal mutations. We also expected that the two-cell embryo microinjection approach might generate adult chimeric F0 mice with germline transmission of lethal mutants. First, we investigated whether the two-cell embryo microinjection method could produce chimeric mice with embryonically lethal mutations. The two-cell embryos, of which one blastomere was microinjected with Cas9 mRNA and Virma sgRNA1 or sgRNA2, were transplanted into the oviducts of pseudopregnant females to obtain chimeric mice. About 10% of the newborn mice clearly carried frameshift mutations (Fig. 1b, c; Table 1). The F0

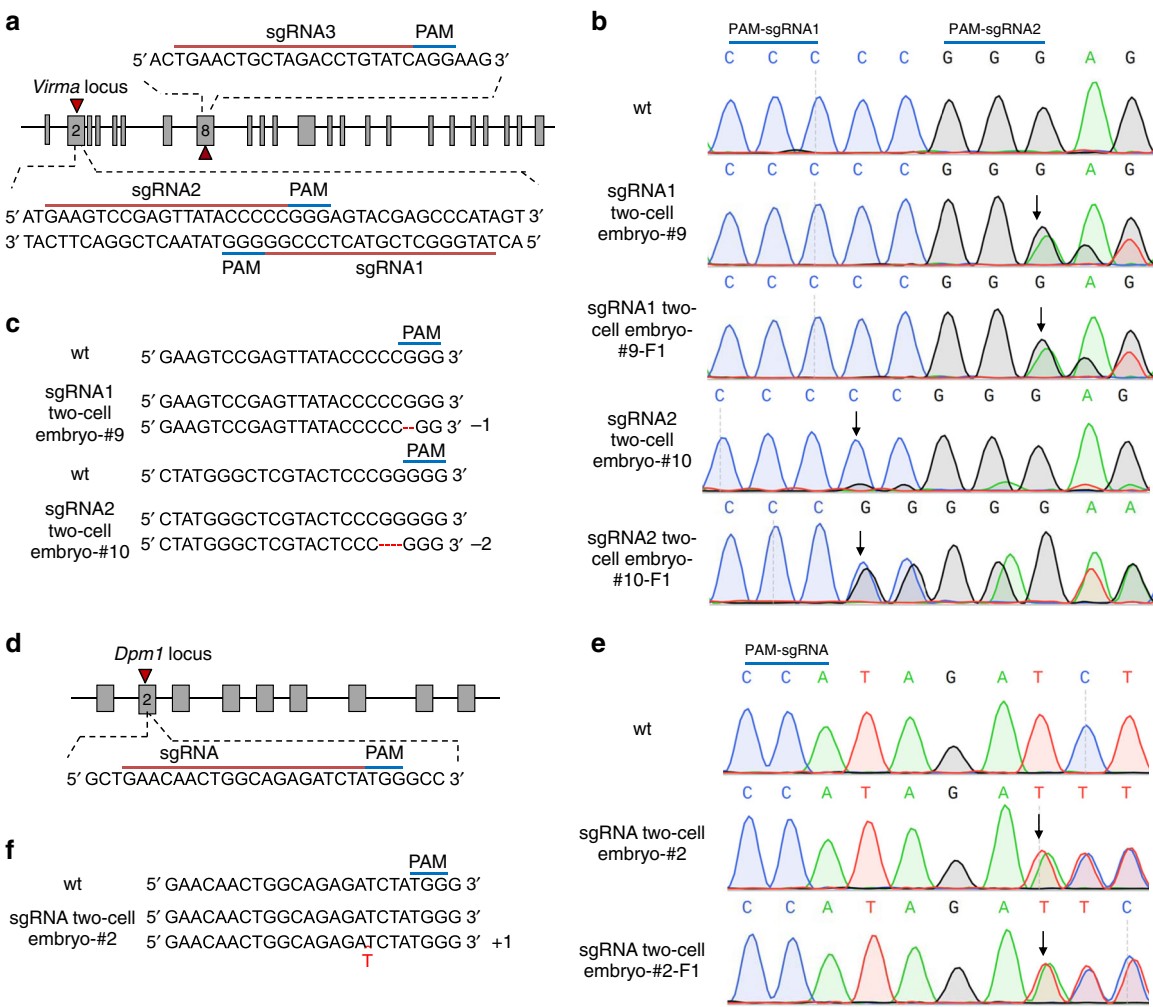

**Fig. 1** Generation of heritable founder (F0) mice with *Virma* or *Dpm1* insertion/deletion mutations. **a** Schematic diagram of sgRNA targeting sites in the mouse *Virma* locus. PAM = protospacer adjacent motif. **b** Sequencing traces of PCR products encompassing the *Virma* target region from wild-type (wt) mice and from representative mutant F0 mice (#9 and #10), generated by microinjecting Cas9 mRNA and sgRNA1 or sgRNA2 into one blastomere of two-cell embryos, and their filial generation (F1). Overlapping sequencing traces among the mutant mice indicate the presence of more than one allele among these mice, compared with wt mice. The mutations' start positions are indicated by black arrows. **c** PCR sequences from representative mutant mice generated by microinjecting different sgRNA targeting sites into one blastomere of two-cell stage embryos. PCR products were analyzed with Sanger sequencing. Deleted nucleotides are indicated by red hyphens. **d** Schematic diagram of sgRNA targeting site in the mouse *Dpm1* locus. **e** Sequencing traces of PCR products encompassing the *Dpm1* target region from wt mice and from a representative mutant F0 mouse and one of its offspring (F1). Overlapping sequencing traces among the mutant mice indicate the presence of more than one allele among these mice, compared with wt mice. The mutations' start positions are indicated by black arrows. **f** PCR sequences from wt and a representative mutant mouse produced by microinjection into one blastomere of a two-cell stage embryo. PCR products were analyzed with Sanger sequencing. The mutated base is labeled in red

mice carrying frameshift mutations survived for more than 120 days.

Successful germline transmission of lethal mutations is essential for establishing genetically modified mouse models. To test the transmissibility of the lethal mutation induced by two-cell embryo microinjection with Cas9 mRNA and sgRNA, *Virma* mutant F0 mice were crossed with wild-type mice. The cross results showed that the *Virma* mutation was effectively transmitted to the F1 generation, regardless of whether sgRNA1 (3/17, 17.6%) or sgRNA2 (3/10, 30%) was used (Fig. 1b; Table 2). Moreover, none of the 34 offspring (F2) of the F1 crossing (*Virma*$^{+/-}$ × *Virma*$^{+/-}$) were homozygous (Supplementary Table 2), thus confirming that *Virma* is embryonically lethal when it is knocked out in mice.

To validate the OSTCM method for producing adult F0 mice with an embryonically lethal knockout phenotype, we chose to target *Dpm1*[2]. Dolichyl-phosphate mannosyltransferase subunit 1

(DPM1), the committed-step enzyme in the N-glycosylation pathway, is tethered to and stabilized on the endoplasmic reticulum membrane[11]. The results of using our method showed that about 27.8% of resulting newborn mice clearly carried *Dpm1* frameshift mutations (Fig. 1d–f; Table 1). Germline transmission of mutations was also found in surviving *Dpm1* mutant mice (Fig. 1e; Table 2).

**F0 mice with heritable early postnatally lethal mutations**. We also used our simplified OSTCM method to produce adult F0 mice carrying early postnatally lethal frameshift mutations. First, we tested whether adult F0 mice with early postnatally lethal mutations could be produced by the traditional method of zygote microinjection. We targeted *Slc17a5*, a gene that causes early postnatal death[12], for zygote microinjection with Cas9 mRNA and *Slc17a5* sgRNA (Fig. 2a). The four F0 mice (Table 1) carrying the frameshift insertion/deletion mutations (Fig. 2b, c) all displayed severe tremors and uncoordinated gaits, appeared

**Table 1 Results of CRISPR/Cas9-mediated targeting in C57BL/6J mice**

| Gene | Group | Embryos transferred (recipients) (n) | Newborns (n) | Mutant mice (n) | NHEJ efficiency (%) | Mice carrying frameshift mutations (n) | Mice carrying a clear frameshift mutation (n) | Surviving adult mutant mice (n) |
|---|---|---|---|---|---|---|---|---|
| Virma | sgRNA1 zygote | 216 (10) | 0 | – | – | – | – | – |
| | sgRNA1 two-cell embryo | 70 (2) | 10 | 1 | 10 | 1 | 1 | 1 |
| | sgRNA2 zygote | 160 (8) | 0 | – | – | – | – | – |
| | sgRNA2 two-cell embryo | 110 (3) | 20 | 2 | 10 | 2 | 2 | 2 |
| | sgRNA3 zygote | 98 (4) | 0 | – | – | – | – | – |
| Dpm1 | sgRNA two-cell embryo | 128 (4) | 25 | 7 | 28 | 7 | 7 | 6 |
| Slc17a5 | sgRNA zygote | 30 (1) | 5 | 5 | 100 | 4 | 4 | 0 |
| | sgRNA two-cell embryo | 51 (2) | 18 | 10 | 55.6 | 7 | 4 | 7 |
| Ctla-4 | sgRNA two-cell embryo | 41 (2) | 12 | 10 | 83.3 | 9 | 3 | 6 |

NHEJ nonhomologous end joining

**Table 2 Summary of germline transmission of lethal mutations produced using the OSTCM method**

| Gene | Group | Assayed mutant F0 mice | Indel Frequencies (%) | New F1 mice (n) | Mutant F1 mice (n) | F1 mice carrying frameshift mutations (n) | Rate of germline transmission of lethal mutation (%) |
|---|---|---|---|---|---|---|---|
| Virma | sgRNA1 two-cell embryo | #9 | 32.8 | 17 | 3 | 3 | 17.6 |
| | sgRNA2 two-cell embryo | #10 | 7.5 | 10 | 3 | 3 | 30 |
| Dpm1 | sgRNA two-cell embryo | #2 | 42.9 | 9 | 3 | 3 | 33.3 |
| | | #10 | 15.5 | 8 | 1 | 1 | 12.5 |
| | | #18 | 45.0 | 9 | 3 | 3 | 33.3 |
| | | #19 | 35.0 | 7 | 4 | 4 | 57.1 |
| Slc17a5 | sgRNA two-cell embryo | #7 | 45.5 | 4 | 3 | 3 | 75.0 |
| | | #11 | 17.4 | 10 | 0 | 0 | 0 |
| | | #12 | 39.7 | 8 | 6 | 2 | 25.0 |
| | | #17 | 15.4 | 8 | 1 | 1 | 12.5 |
| | | #18 | 12.6 | 5 | 1 | 0 | 0 |
| Ctla-4 | sgRNA two-cell embryo | #10 | 47.2 | 7 | 3 | 3 | 42.9 |

Roughly 25% of mouse genes are embryonically lethal when knocked out, preventing the generation of viable mouse models. Here, the authors use CRISPR-Cas9 to edit one blastomere of a two-cell embryo to generate viable chimeric mice

weak, and typically died during the third postnatal week (Supplementary Movie 1), thus limiting transmission of their frameshift mutations. Their phenotypes were consistent with the previously described characterization of a $Slc17a5^{-/-}$ mouse in its third postnatal week[12].

We then investigated whether OSTCM could produce chimeric mice with early postnatally lethal mutations. The two-cell stage embryos, of which one blastomere was microinjected with Cas9 mRNA and *Slc17a5* sgRNA (Fig. 2a), were transplanted into the oviducts of pseudopregnant females to obtain chimeric mice. About 55.6% of the newborn pups clearly carried mutant sequences (Fig. 2b, c; Table 1). However, the percentage of surviving adult mice carrying frameshift alleles generated by the one-blastomere microinjection method rose to 100%, compared with 0% in the zygote microinjection group (Table 1). The F0 mice carrying frameshift mutations produced by our OSTCM method survived for more than 120 days.

To test the transmissibility of an early postnatally lethal mutation induced by the OSTCM method, we crossed *Slc17a5* mutant F0 mice with wild-type mice. The cross results showed that the *Slc17a5* mutations were effectively transmitted to the subsequent (F1) generation (Fig. 2d; Table 2).

To validate our OSTCM method for producing adult F0 mice with a postnatally lethal knockout phenotype, we targeted the cytotoxic T-lymphocyte antigen-4 (*Ctla-4*) gene, a gene that causes early postnatal death[13]. Our results here showed that newborn mice carrying the *Ctla-4* frameshift mutation can also be obtained using the OSTCM method (Fig. 2e–g, Table 1). Although three of the nine F0 mice carrying frameshift mutations died at around 21 days after birth, the others survived for more than 150 days (Table 1). We also saw germline transmission of mutations in surviving *Ctla-4* mutant mice (Table 2). Clearly, the early postnatally lethal mutations induced by the OSTCM method can be transmitted to the next generation.

We performed PCR amplifications of three targeted loci: *Virma* (Fig. 3a), *Slc17a5* (Fig. 3b), and *Ctla-4* (Fig. 3c) from different tissues (tail, brain, liver, heart, lung, spleen, salivary glands, and regulatory T (Treg) cells). Sanger sequencing analysis showed that

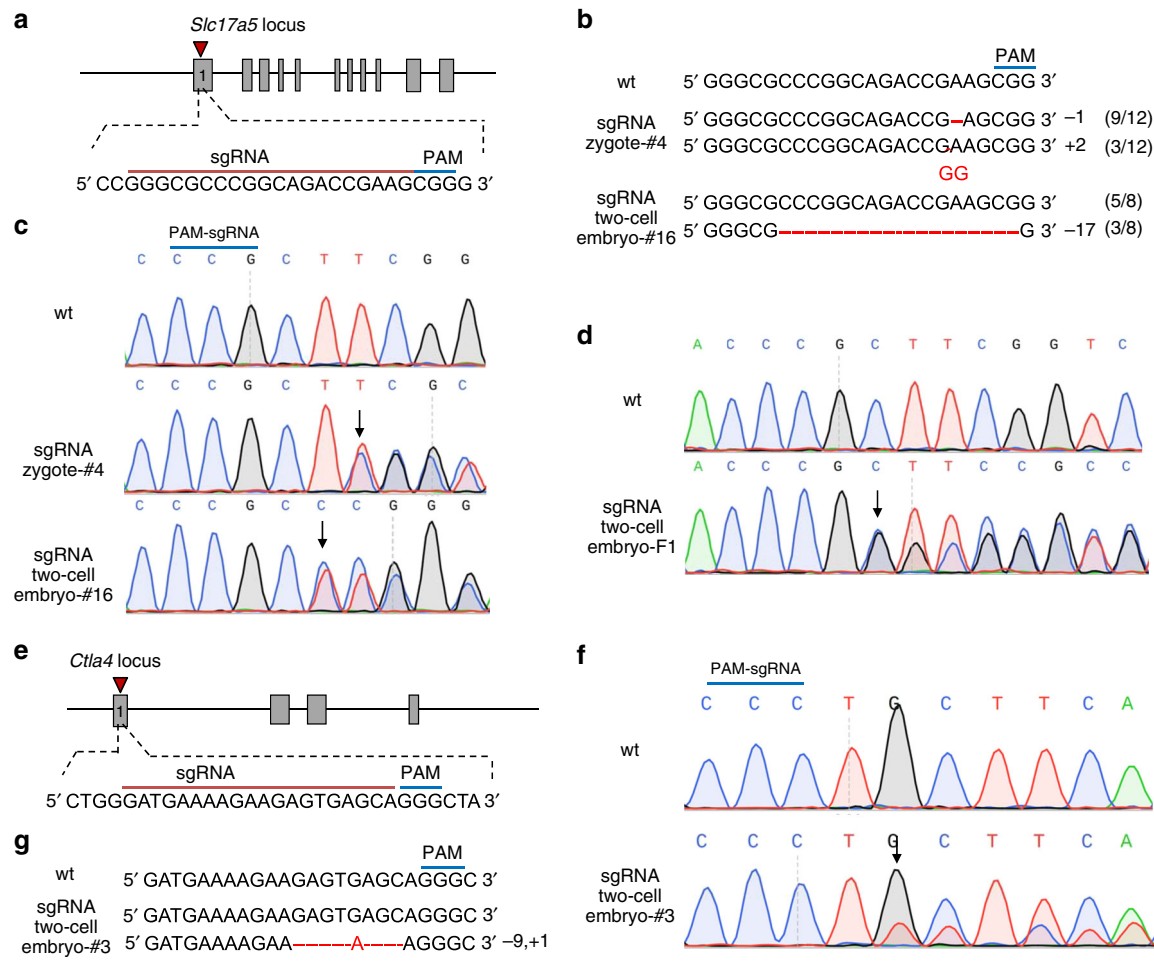

**Fig. 2** Heritable F0 mice with *Slc17a5* or *Ctla-4* insertion/deletion mutations. **a** Schematic diagram of the sgRNA targeting site in the mouse *Slc17a5* locus. PAM = protospacer adjacent motif. **b** PCR sequences encompassing the *Slc17a5* target region from wild-type (wt) mice and from representative mutant mice (#4 and #16) generated by two methods of microinjection: (1) into zygotes using sgRNA; and (2) into one blastomere of two-cell stage embryos using sgRNA. PCR products were sequenced and those showing overlapping sequencing traces were cloned. The subsequent individual clones were then sequenced. Mutated bases are labeled in red and deleted nucleotides are indicated by hyphens. Numbers and percentages of the mutated bases are listed on the right. **c** Sequencing traces of PCR products from representative mutant mice generated by the aforementioned two microinjection methods. Overlapping sequencing traces among the mutant mice indicate the presence of more than one allele among these mice, compared with wt mice. The mutations' start positions are indicated by black arrows. **d** Germline transmission of the *Slc17a5* lethal mutations in F1 progeny is shown by sequencing traces of PCR products encompassing the *Slc17a5* target region from a representative mutant F1 mouse. Overlapping sequencing traces indicate heterozygous mice. The black arrow shows the mutation starting position. **e** Schematic diagram of the sgRNA targeting site in the mouse *Ctla-4* locus. **f** Sequencing traces of PCR products encompassing the *Ctla-4* target region from wt mice and from a representative mutant F0 mouse. The mutant mouse's overlapping sequencing traces indicate the presence of more than one allele, compared with wt mice. The mutation start position is indicated by a black arrow. **g** PCR sequences of the *Ctla-4* target region of wt mice and a representative mutant mouse generated by microinjection into one blastomere of two-cell embryos. The mutated base is labeled in red and deleted nucleotides are indicated by hyphens

the mutation pattern found in the tails of founders generated by our OSTCM method is the same throughout the animal and that editing efficiency, as assessed by TIDE (tracking of indels by decomposition) analysis[14], is homogeneous in each tissue. Therefore, genotyping results from mouse tails accurately reflect the mutant state of chimeric mice, thus allowing us to easily choose the mice that carry one or two mutataions in tissues or cells for further analysis.

**F0 mice with lethal mutations produced by IVF.** Optimization and simplification of microinjection conditions will facilitate the potential applications of our OSTCM method. To increase the overall efficiency of generating lethal mutants, we incorporated the Cas9 ribonucleoprotein complex and in vitro fertilization (IVF) technology into the OSTCM method. When the IVF

zygotes developed into two-cell embryos (cultured in vitro for 12 h), we microinjected Cas9 ribonucleoprotein complexes of *Slc17a5* or *Ctla-4* into one blastomere of two-cell embryo. About 40% (*Slc17a5*) and 80% (*Ctla-4*) of the resulting newborn pups clearly carried one or two mutant sequences (Supplementary Table 3). This suggested that our OSTCM method, combined with Cas9 ribonucleoprotein complex and IVF, can be used to produce mice with lethal mutations, although its knockout efficiency was not better than that of using Cas9 mRNA in combination with in vivo fertilization.

**CRISPR off-target analysis in F0 mice.** Recent studies in genetically engineered rats and mice using the zygote micro-injection method have suggested that there is a certain level of off-target cleavage by the CRISPR/Cas system[15], indicating that

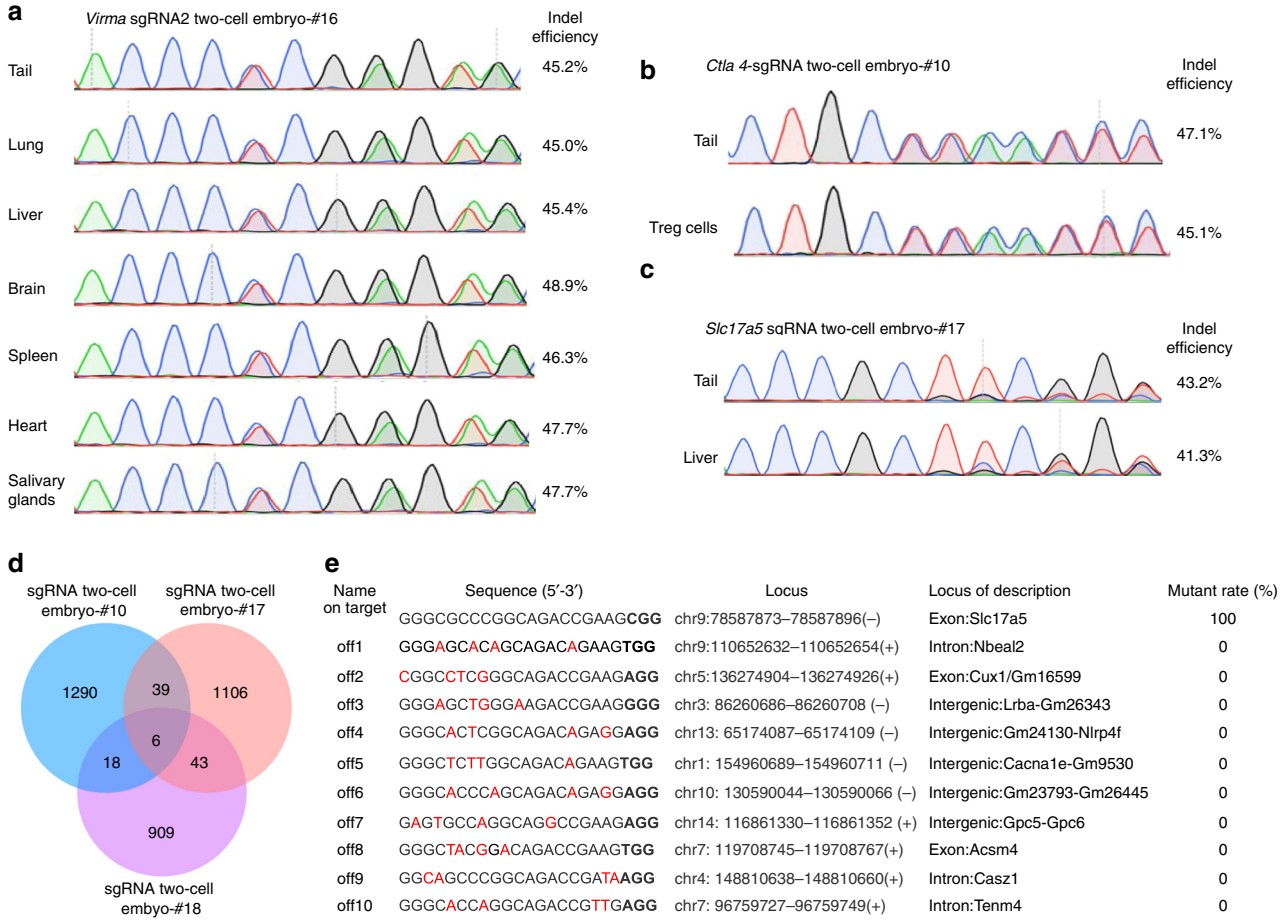

**Fig. 3** Off-targets identified from *Slc17a5* F0 mice. Sequencing traces of PCR products encompassing the target regions of *Virma* **a**, *Ctla-4* **b**, or *Slc17a5* **c** in the tissues or Treg cells of representative mutant mice produced by the OSTCM method. Overlapping sequencing traces indicate the presence of more than one allele among the mutant mice. Editing (indel) efficiencies assessed by TIDE analysis in each tissue is shown to the right of each trace. **d** Venn diagram displaying the overlap of all single-nucleotide variants detected in whole-genome sequencing data from three mutant mice (#10, #17, and #18). **e** The top ten potential off-target sites of CRISPR-Cas9 for sgRNA. Red nucleotides represent a mismatch compared with the on-target sequences. Ten selected potential off-target sites analyzed with Sanger sequencing of the livers of the three mutant mice from **d**

Cas9-mediated DNA cleavage tolerates small numbers of mismatches between sgRNA and target DNA, especially in the protospacer adjacent motif-distal region. For that reason, we tested for possible off-target effects in genome-modified mice derived by the OSTCM method. We used whole-genome sequencing data from three *Slc17a5* mutant mice (#10, #17, and #18) and three wild-type mice to analyze systematic off-target effects. With this analysis, we found six common off-targets among three mutant mice (Fig. 3d; Supplementary Fig. 2). However, these six off-targets were not located in protein coding regions in the mouse genome (Supplementary Table 4). Then we used the guide RNA selection tool CRISPOR[16] to find the top ten potential off-target sites for sgRNA (Fig. 3e), and then we analyzed potential off-target effects by Sanger sequencing in the livers of the three mutant mice. None of the sequencing reads had mutations, suggesting there were no off-target effects at these sites (Fig. 3e).

**_Virma_ knockout leads to focal segmental glomerulosclerosis.** m6A is enriched in the 3′ untranslated region and near the stop codon of mature polyadenylated mammalian mRNAs. It has regulatory roles in a eukaryotic mRNA transcriptome switch. However, since *Virma* mutations cause embryonic lethality, *Virma* knockout mice have been, by extension, unavailable. Therefore, previous studies were mostly carried out in culture[9,10],

making it difficult to explore the effects of this mutation on in vivo functioning.

Here, we studied VIRMA's functions in F0 chimeric mice with lethal mutations. In situ hybridization (ISH) results demonstrated that *Virma* mRNA was present in the kidneys of wild-type mice, but that its expression was significantly decreased in the F0 knockout mice generated using our OSTCM method (Fig. 4a). q-PCR using a specific primer-targeting *Virma* mutant region demonstrated that *Virma* mRNA expression in kidney tissues of F0 mice generated by our OSTCM method decreased by about 50% compared with wt mice (Fig. 4b). Individual clones were sequenced from RT-PCR products, and the results showed that frameshift mutations, that completely knockout gene function, were present in about 50% of *Virma* mRNA in kidney tissue from knockout F0 mice (Fig. 4c). We compared the histopathological changes in various organs (kidney, lung, liver, submandibular gland, brain, spleen, heart, kidney, and thymus) of Virma mutant mice, generated by OSTCM, with wild-type mice (Supplementary Fig. 3). Hematoxylin/erythrosine (HE) staining results showed that the diameter of the knockout animals' glomeruli was significantly larger than that of the wt group's (Fig. 4a; Supplementary Fig. 3). Biopsies of both wt and knockout animals' kidneys revealed complement component C3 and IgM deposition in the knockout animals' glomeruli (Fig. 4d, e). We analyzed differentially expressed genes using RNA-seq of kidney tissue

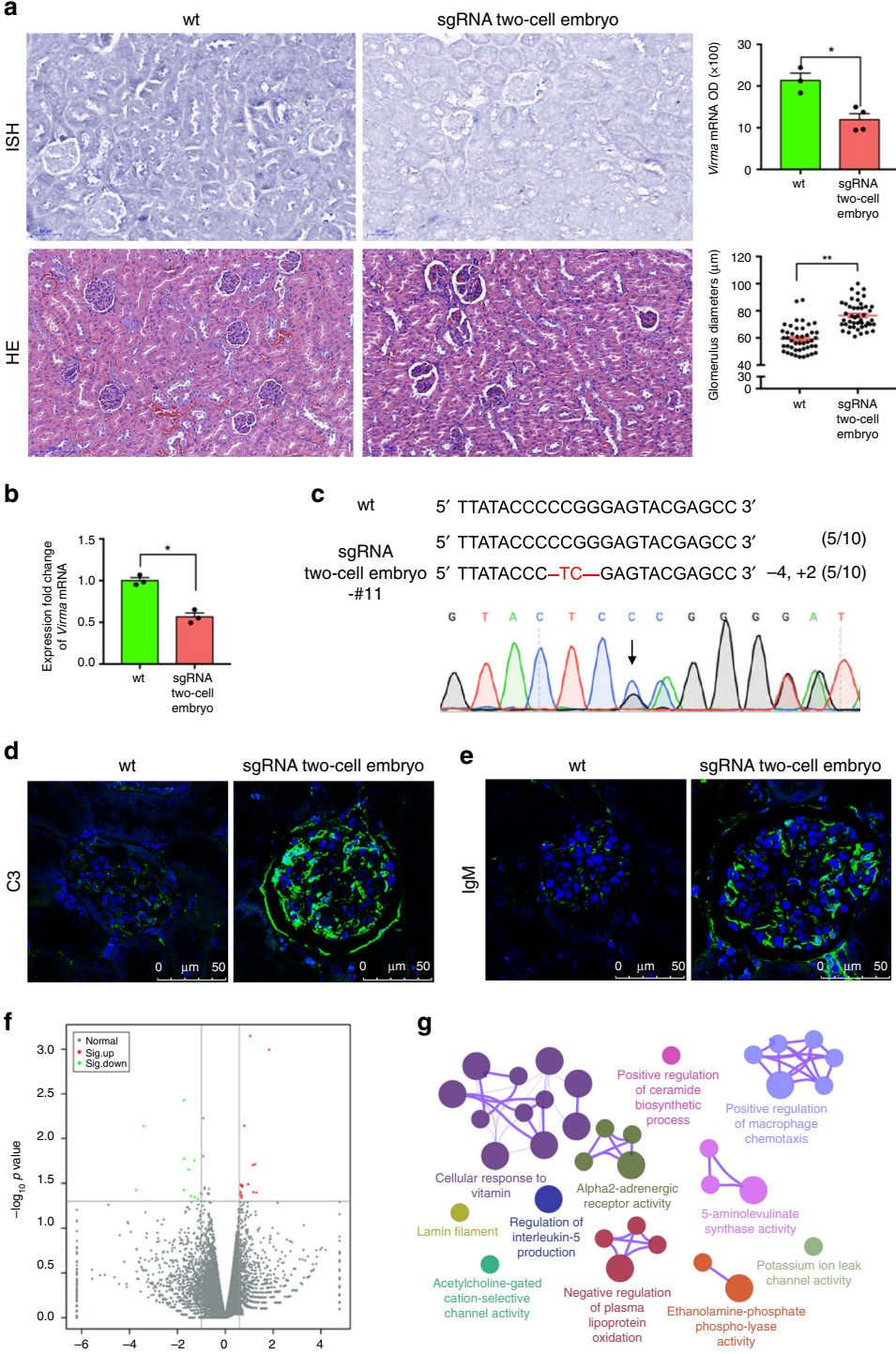

**Fig. 4** *Virma* knockout led to developmental glomerulus abnormalities in adult mouse kidneys. **a** Two images, top left: in situ hybridization (ISH) of *Virma* mRNA in the kidneys of wild-type (wt) mice and of *Virma* knockout F0 mice generated by the OSTCM method, scale bar: 100 μm; bar chart, top right: quantification of mRNA; two images, lower left: hematoxylin/erythrosine (HE) stained kidney glomeruli from wt mice and from *Virma* knockout F0 mice, scale bar: 100 μm; lower right: glomerulus diameters. **b** q-PCR results of the differential expression of *Virma* mRNA in kidney tissue between wt mice and knockout F0 mice. **c** RT-PCR products of *Virma* mRNA from the kidney tissue of knockout F0 mice were cloned and sequenced. Upper panel: mutated bases are labeled in red and deleted nucleotides are indicated by hyphens. Numbers and percentages of the mutated bases are listed on the right. Lower panel: sequencing traces of RT-PCR products. The mutant mouse's overlapping sequence traces indicate the presence of more than one allele. The mutation start position is indicated by a black arrow. **d**, **e** Confocal microscopy of C3 (**d**) or IgM (**e**) deposits in the kidney glomeruli in adult *Virma* knockout F0 mice. DAPI (blue), C3 protein or IgM protein (green). Scale bar: 50 μm. **f** Differentially expressed genes (fold change > 1.5 or < 0.5) identified by RNA-seq in kidney tissue from both wt mice and *Virma* mutant mice. **g** Gene ontology (GO) and enrichment analysis of the differentially expressed genes. Values are means ± SEM. Asterisks in all plots, *P*-values after unpaired *t* test, two-tailed, * indicates significance at *P* < 0.05, ** indicates significance at *P* < 0.01. Source data are provided as a Source Data file

from both wt mice and *Virma* mutant mice generated by the OSTCM method. Based on two replicate samples, there are 16 upregulated and 11 downregulated genes with a greater than 1.5-fold expression change (Fig. 4f; Supplementary Table 5). The differences in *Virma* mRNA expression (Supplementary Table 5) and coverage of mapped reads (Supplementary Fig. 4) failed to be detected between wt mice and knockout F0 mice using RNA-seq technology. It was suggested that the *Virma* transcripts containing premature termination codons generated by frameshift mutations were not degraded efficiently[17]. Gene ontology analysis of differentially expressed genes yielded an enrichment network of focal segmental glomerulosclerosis (FSGS) (Fig. 4g), in which notable gene clusters include cellular responses to vitamin and positive regulation of macrophage chemotaxis. This indicates that VIRMA protein regulates FSGS in adult mouse kidneys.

**Slc17a5 gene-regulated sialic acid and nitrate secretions**. *Slc17a5* encodes for sialin, which acts as a proton-driven carrier for sialic acid across lysosomal membranes[12], as a nitrate transporter in the plasma membrane[18], and as a vesicular glutamic acid-aspartate cotransporter in the brain[19]. However, as *Slc17a5* knockout mice die prematurely[12], previous studies were mostly carried out in culture, using short hairpin RNA (shRNA)-mediated knockdown, or in a knockout mouse model before 18 days after birth, leaving the effects of this mutation on in vivo adult functioning relatively unexplored. Here, we studied the functions of sialin in acinar cells of adult submandibular glands using F0 chimeric mice with lethal mutations.

Immunohistochemistry (IHC) and ISH results demonstrated that *Slc17a5* mRNA and sialin existed mostly in acinar cells in wt mice. When *Slc17a5* was knocked out in one part of acinar cells, the rest of the cells showed significantly decreased expression of *Slc17a5* mRNA and sialin in F0 mice generated using our OSTCM method (Fig. 5a). q-PCR results demonstrated a 70% decrease in *Slc17a5* mRNA expression in submandibular glands of F0 mice generated by our OSTCM method compared with wt mice (Fig. 5b). *Slc17a5* knockout downregulated transmembrane protein 16 A (TMEM16A) and aquaporin 5 (AQP5) expressions in the acinar cells of F0 knockout mice generated using the OSTCM method (Fig. 5c).

Water-immersion-restraint stress resulted in decreased secretion of sialic acid in the saliva of F0 knockout mice generated using OSTCM method (Fig. 5d). After water-immersion-restraint stress, $NO_3^-$ secretion also decreased in the saliva, but not in the serum, of F0 knockout mice (Fig. 5e). The calcium-activated chloride channel TMEM16A contributes to transepithelial chloride transport[20], and AQP5 plays a fundamental role in transmembrane water movement in animal tissues[21]. Therefore, sialin may be involved in acinar cells' sialic acid and nitrate secretions through TMEM16A and AQP5 proteins' actions in adult submandibular glands.

**Ctla-4 knockout leads to Treg cell proliferation**. CTLA-4 is vital for regulating T cell-mediated suppression. Blocking the inhibitory effects of CTLA-4 allows for effective immune responses against tumor cells[22]. For example, anti-CTLA-4 antibody treatment enhances tumor immunity in metastatic melanoma patients[23]. In mice, a conventional *Ctla-4* knockout causes lymphocytic infiltration and the destruction of major organs, and is lethal 3–4 weeks after birth[13]. Conditional *Ctla-4* knockout mice produced by crossing CD4[cre] mice with *Ctla-4*[fl/fl] mice die at around 21 days of age due to a wasting disease[24]. This early postnatal lethality prohibits the study of *Ctla-4*'s specific role in central and peripheral tolerance and in systemic diseases in adulthood. The inducible deletion of *Ctla-4* in adult mice

increased the frequency of Foxp3(+) Treg cells in the spleen[24]. Similarly, the frequencies of CD25(+) CD127(−) or Foxp3(+) Treg cells were significantly elevated in the spleens of F0 knockout adult mice generated using OSTCM method (Fig. 6a). We isolated a single CD3(+)CD4(+)CD25(+)CD127(−) Treg cell from spleen tissue of one F0 knockout mouse and performed single-cell PCR, followed by sequencing analysis. The results showed that the OSTCM method led to both the same bi-allelic editing in targeted cells as in tail tissue (Fig. 6b) and a reduction of CTLA-4 levels in CD3(+)CD4(+)CD25(+)CD127(−) Treg cells, regardless of whether the in vitro treatment (aCD3: 3 μg/mL, aCD28: 2 μg/mL, and IL2: 20 ng/mL for 3 days) was used (Fig. 6c, d). This indicates that CTLA-4 protein regulates Treg cell activation in adult mouse spleens.

**Discussion**

Here, we have shown that microinjection of the CRISPR-Cas9 system into one blastomere of two-cell embryos can, in a single step, efficiently generate adult mice carrying heritable lethal mutations, thus enabling rapid establishment of a conventional knockout animal model for a lethal mutation. Regardless of whether the gene of interest was embryonically or postnatally lethal, frameshift mutations were transmitted to the F1 generation by crossing with wt mice. Consequently, the homozygous off-spring could be used to study gene function in vivo during their embryonic or early postnatal stages before their deaths. In some instances, genes have different functions in adults than in developing embryos or newborns. By examining surviving F0 mice with lethal mutations, the phenotype and function of these genes can be identified in adult animals (Fig. 7).

The CRISPR-Cas9 gene editing system has been widely used to rapidly generate knockout mice by microinjecting zygotes[4]. However, this method is unable to generate either F0 newborns with embryonically lethal mutations or with postnatally lethal mutations. Our study results demonstrate that F0 newborns could not be produced by zygote microinjection, using the CRISPR-Cas9 system with different sgRNA sequence features, to knock out *Virma* (leading to embryonic lethality) and that F0 newborns produced by zygote microinjection using the CRISPR-Cas9 system to knock out *Slc17a5* (leading to early postnatal lethality) failed to grow into adulthood. The lack of adult F0 mice with lethal phenotypes using the CRISPR-Cas9 system by zygote microinjection limits the generation of conventional knockout animal models. Also, the traditional method for targeting genes in embryonic stem cells to generate conventional knockout mice with lethal phenotypes[25,26] is complex, labor intensive, and time consuming, taking 6–12 months or longer[27].

When working with embryonic or postnatal lethality using conventional knockout mouse models, gene function can be assessed only during the early developmental or premature stages. Knockout mice with lethal mutations must survive longer than that in order to study gene function in vivo in the postnatal or adult stages. This problem is usually circumvented by using a method based on the bacteriophage P1-derived Cre/*loxP* recombination system for conditional knockouts of endogenous genes[28]. This method involves crossing the floxed mouse strain with a Cre transgenic line under a specific promoter, and usually requires at least two generations to obtain progeny of interest, thus making it yet another time-consuming way to obtain conditional mutant mice. Moreover, this technology cannot conveniently—and is therefore not recommended to—screen gene functions because each floxed strain must be individually crossed with several hundred Cre driver mouse lines.

Recently, the functions of *Tet3*, which regulates excitatory and inhibitory synaptic transmission in the developing mouse

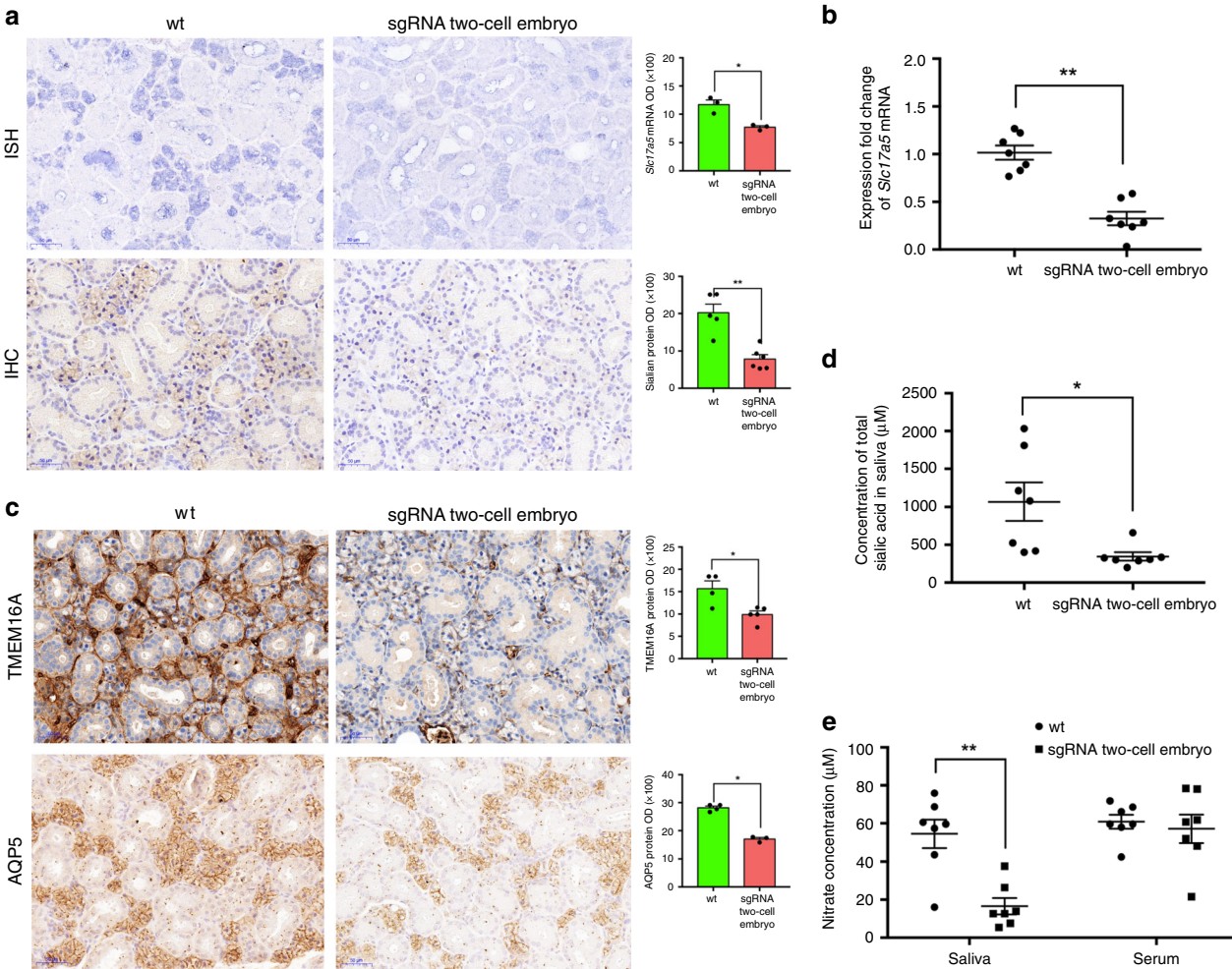

**Fig. 5** *Slc17a5* knockout diminished sialic acid and nitrate secretory functions. **a** In situ hybridization (ISH) and immunohistochemistry (IHC) results for the presence of *Slc17a5* mRNA and sialin protein in tissue slices of submandibular glands of wt mice and *Slc17a5* knockout F0 mice generated by the OSTCM method. Quantifications of mRNA and protein are shown in panels on the right. Scale bar: 50 μm. **b** q-PCR results show differing expressions of *Slc17a5* mRNA in submandibular glands between wt mice and *Slc17a5* knockout F0 mice. **c** IHC (left panels) and quantification (right panels) of transmembrane protein 16 A (TMEM16A) and aquaporin 5 protein (AQP5) in submandibular glands of wt mice and of *Slc17a5* knockout F0 mice. Scale bar: 50 μm. **d** Comparison of total sialic acid in saliva, after water-immersion-restraint stress of wt mice and *Slc17a5* knockout F0 mice. **e** Comparison of $NO_3^-$ in saliva and serum after water-immersion-restraint stress of both wt mice and *Slc17a5* knockout F0 mice. Values are means ± SEM. Asterisks in all plots, *P*-values after unpaired *t* test, two-tailed, * indicates significance at $P < 0.05$, ** indicates significance at $P < 0.01$. Source data are provided as a Source Data file

cerebral cortex, were discovered in adult mice produced by a two-step microinjection method that generated chimeric mutant mice with a postnatally lethal knockout phenotype[5]. Although this method allows for in vivo analysis of the functions of postnatally lethal knockout genes, it does have limitations: It cannot be used in general mouse lines, its microinjection procedures are complex, and it has not been tested to see if it can produce knockout mice with embryonically lethal mutations. In order to increase the percentage (∽50%) of KO cells in mice, this two-step microinjection method requires an initial injection of Cas9 mRNA during the zygote stage, followed by a second injection of sgRNA and Cre mRNA into one blastomere of the resulting two-cell embryo. However, the high percentage of lethally mutant cells in embryos might fail to produce knockout F0 mice. With our one-step microjection method, the efficiency of producing viable F0 mice with embryonically lethal mutations (*Virma* or *Dpm1*) was obviously lower than that of producing mice with postnatally lethal knockouts (*Slc17a5* or *Ctla-4*), but it did produce viable F0 mice.

Our one-step method of microinjecting two-cell embryos, using CRISPR/Cas9-mediated mutagenesis that simplifies the manipulation procedure, is independent of sgRNA sequence features, and can be applied in in vivo investigations of genes with embryonically or postnatally lethal knockout phenotypes in general mouse lines. Using our OSTCM method, a conventional knockout animal model with a lethal phenotype may be established in only 3 months. This method also allows rapid screening of multiple tissues and cell types, thus eliminating the time required to produce at least two generations of crosses by using multiple drivers in the Cre/*loxP* system. A much larger number of lethal genes than those targeted in this study most likely can be modified by using this method. However, whole-genome sequencing to test off-target effects induced by CRISPR-Cas9 must be done before using this approach to analyze gene function in vivo. Also, variability in the degree of mosaicism within the embryos should be considered when analyzing in vivo gene functions. To overcome these potential problems, chimeric mice with only one or two types of a mutant gene, given similar editing efficiency measured by genotyping and by TIDE analysis of tail

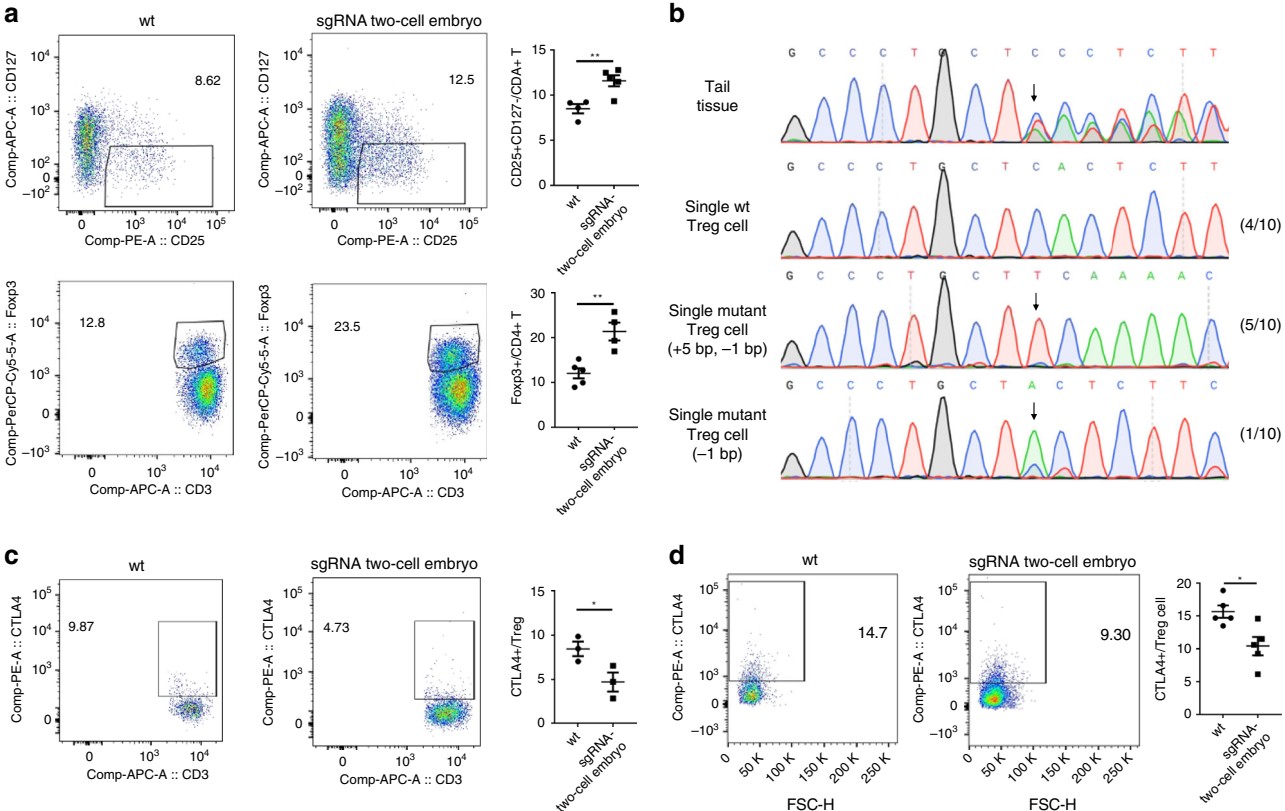

**Fig. 6** Activation of Treg cells by *Ctla-4* knockout in adulthood. **a** Frequencies of CD25(+)CD127(−) (Up) and Foxp3(+) (Down) Treg cells in spleens of *Ctla-4* knockout and wt mice. Representative fluorescence-activated cell sorting (FACS) results are in the left panels, and the corresponding statistical results are in the right panel. *n* = 4–5 mice per group. **b** Single-cell genotyping of a single CD3(+)CD4(+)CD25(+)CD127(−) Treg cell from the spleen of one mutant chimera. The cell was PCR amplified and Sanger sequenced. Sequencing traces of PCR products from representative wt and mutant cells generated by the OSTCM method are shown. The mutations' start positions are indicated by black arrows. Numbers and percentages of the mutated cells are listed on the right. **c**, **d** Frequencies of CTLA-4(+) cells among CD3(+)CD4(+)CD25(+)CD127(−) Treg cells in the spleens of *Ctla-4* knockout and wt mice before (**c**) and after (**d**) in vitro treatment (aCD3: 3 μg/mL, aCD28: 2 μg/mL, and IL2: 20 ng/mL for 3 days). Representative FACS results are in the left panels, and the corresponding statistical results are in the right panels. *n* = 3–5 mice per group. Values are means ± SEM. Asterisks in all plots, *P*-values after unpaired *t* test, two-tailed, * indicates significance at *P* < 0.05, ** indicates significance at *P* < 0.01. Source data are provided as a Source Data file

DNA, may be used for further functional analyses. Meanwhile, a potential strategy to analyze the different phenotypes between wild-type and knockout cells, taken from the same tissue from embryos produced by OSTCM, is to use a combined method of single-cell PCR and single-cell RNA sequencing. Moreover, knock-in efficiencies achieved by introducing CRISPR reagents into embryos at the two-cell stage are greater than tenfold those achieved at the zygote stage[29]. The higher frequencies of off-target single-nucleotide variants induced by cytosine base editing was 20-fold higher than those produced by CRISPR/Cas9, according to a genome-wide off-target analysis of the one-blastomere two-cell embryo injection method[30]. Our method of using two-cell embryos may have serious potential for producing genetically modified mouse models.

In summary, the OSTCM method is an important addition to the current collection of genetic tools, particularly those aimed at rapid establishment of knockout animal models with lethal phenotypes using the CRISPR-Cas9 system.

## Methods

**Animal ethics statements**. Animal experiments were conducted in accordance with the National Institutes of Health Guide for the Care and Use of Laboratory Animals and all procedures were approved by the Animal Ethics Committee at Capital Medical University (Beijing, China; approval no. AEEI-2017-009). All surgical procedures were performed under chloral hydrate anesthesia, and all efforts were made to minimize animal suffering.

**Animals**. Five-week-old female C57BL/6JCnc (B6J) mice (Cat#: 219; chosen to superovulate oocytes for in vivo or in vitro fertilization), three-month-old male B6J mice (Cat#: 219; chosen for in vivo or in vitro fertilization), and three-month-old female ICR mice (Cat#: 201; chosen as foster mothers) were purchased from Beijing Vital River Laboratory Animal Technology Co., Ltd., for the study.

**Production of Cas9 mRNA and sgRNA**. T7 promoter was added to the Cas9 coding region by PCR amplification using primer Cas9 F and R (Supplementary Table 6). The T7-Cas9 PCR product was gel-purified and used as the template for in vitro transcription using the mMESSAGE mMACHINE T7 ULTRA kit (Invitrogen). T7 promoter was added to the sgRNA template by PCR amplification using the primers shown in Supplementary Table 6. The T7-sgRNA PCR product was gel-purified and used as the template for in vitro transcription using the MEGAshortscript T7 kit (Invitrogen). Both the Cas9 mRNA and sgRNA were purified using the MEGAclear kit (Invitrogen) and eluted in RNase-free water.

**Embryo collection**. In vivo fertilization: B6J female mice (5-weeks old) superovulated by PMSG and hCG were mated to B6J stud males and fertilized zygotes were collected from oviducts[31]. In vitro fertilization: superovulated oocytes from PMSG and hCG treated B6J female mice were collected from oviducts. The oocytes and capacitated sperm were fertilized in vitro in human tubal fluid medium with 5.14 mM calcium[32]. Zygotes at the pronuclei stage from either in vivo or in vitro fertilization were cultured in the M16 medium for 12 h, and then two-cell embryos were used for microinjection.

**Zygote microinjection**. Cas9 mRNA (12.5 ng/μL) and sgRNA (6.5 ng/μL) were microinjected into the cytoplasm of fertilized eggs with well-recognized pronuclei in the M2 medium. The microinjected zygotes were cultured in M16 at 37 °C under 5% $CO_2$ in air for 12 h until the two-cell embryo stage. Two-cell embryos were transferred into the oviducts of pseudopregnant ICR females at 0.5 day postcoitum.

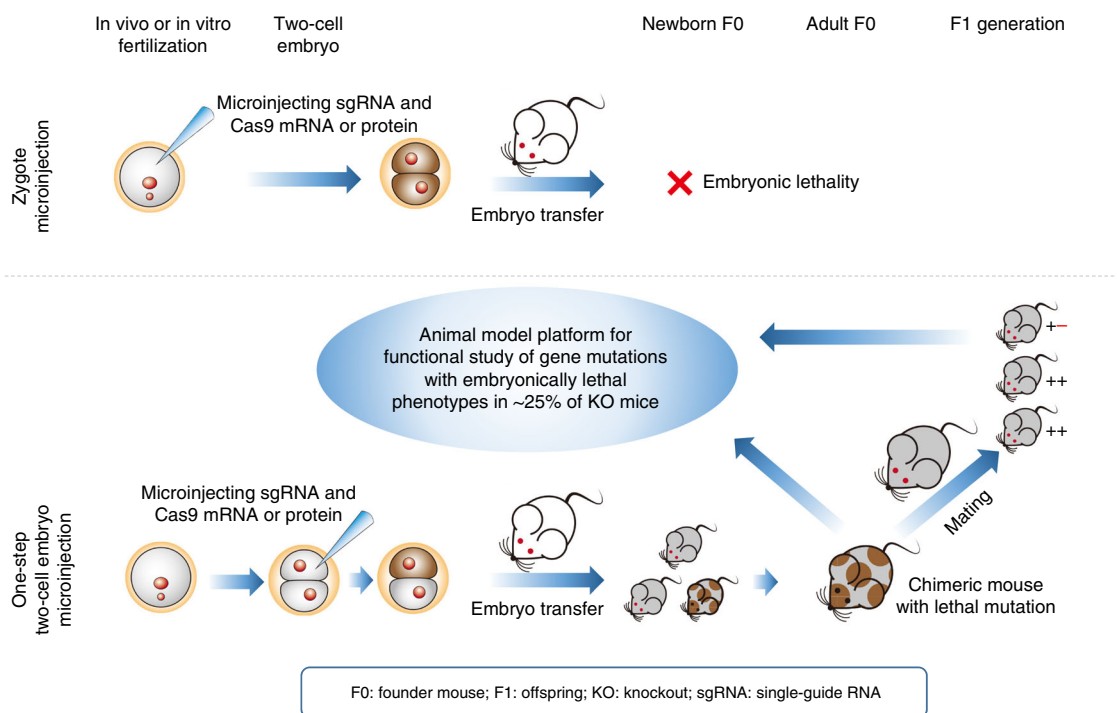

**Fig. 7** Generation of surviving chimeric mice with lethal mutations. Compared with zygote microinjection, microinjecting CRISPR-Cas9 components into one blastomere of two-cell embryos can generate surviving adult chimeric mice with lethal mutations

**Two-cell embryo microinjection**. sgRNA (6.5 ng/μL) and Cas9 mRNA (12.5 ng/μL) or Cas9 protein (5 ng/μL) were microinjected into one blastomere of each two-cell embryo in the M2 medium. The microinjected two-cell embryos were cultured in M16 at 37 °C under 5% $CO_2$ in air for 3 h. Two-cell embryos were then transferred into the oviducts of pseudopregnant ICR females at 0.5 day postcoitum.

**Genotyping of mouse tail DNA**. Mouse pups between 8 and 14 days old were tailed (for DNA) and toed (for identification). The genomic DNA was extracted using the Animal Tissues/Cells Genomic DNA Extraction Kit (Solarbio) and PCR amplified using Hieff[TM] PCR Master Mix (Yeasen). PCR products were directly sequenced after purification or cloned into pEASY-T1 Vector (Transgen) for TA-cloning-based sequencing. The sequencing was carried out by Sangon Corporation. The primers for genotyping are listed in Supplementary Table 6.

**Age of animals used in the gene function study**. Mice (110–120 days old) generated by microinjection of the two-cell embryos and wt mice that were age- and sex-matched were used in the gene function study of Slc17a5, Ctla-4, and Virma.

**Tissue preparation for histological analyses**. Mouse submandibular gland and spleen samples were dehydrated with gradient ethanol and fixed in 4% paraformaldehyde at 4 °C overnight. Samples were then embedded in paraffin and sectioned at 5-μm thickness for later staining.

**ISH**. The total RNA was extracted from submandibular glands or spleens of C57BL/6 mice. The degenerate primers used for Slc17a5 and Virma are listed in Supplementary Table 6. After RT-PCR, the correct-sized bands were extracted from agarose gels, and their DNA sequences were determined. The RNA probe was synthesized using digoxigenin-UTP with T7 RNA polymerase (Roche) according to the manufacturer's protocol (DIG RNA labeling Mix, Roche). For the staining procedure, slides were first rehydrated, then treated with proteinase K (1 μg/ml in PBS) for 30 min at 37 °C, and finally refixed with 4% paraformaldehyde. The fixed specimens were dehydrated in a series of increasing ethanol concentrations (30, 50, 75, 90, and 100%). After drying in air for 1 h, the specimens were hybridized at 70 °C overnight. After 3−4 h of rinsing with SSC solution, specimens were incubated with alkaline phosphatase-conjugated anti-digoxigenin Fab (Roche) overnight. Signals were detected with NBT/BCIP substrates (Promega).

**Immunohistochemistry (IHC)**. Deparaffinized sections were subject to antigen retrieval, followed by treatment with 10% $H_2O_2$/methanol for 10 min to quench the endogenous peroxidase activity. The sections were incubated with primary antibodies at 4 °C overnight. The primary antibodies used in this research were: sialin (1:250, PA5-42456, ThermoFisher), AQP5 (1:200, ab92320, Abcam), TMEM16A (1:40, AF1356, R&D) expression .

**Quantitative real-time RT-PCR of mRNA**. mRNA transcripts were measured using a standard SYBR Green real-time PCR assay. RNA was isolated using RNAprep Pure Tissue Kit (Tiangen), and 1 μg of the total RNA was reverse-transcribed using the RevertAid First Strand cDNA Synthesis Kit (ThermoFisher). Real-time PCR was performed on cDNA in a GoTaq® qPCR Master Mix (Promega) with gene-specific primers (Supplementary Table 6). Gapdh was used as a housekeeping gene. Amplicons were analyzed using the $2^{-\Delta\Delta Ct}$ method, and data are represented as the mean of three independent experiments ± SEM.

**Water-immersion-restraint stress assay**. The water-immersion-restraint stress (WIRS) assay was used in this study to induce gastric mucosal lesions[33]. Mice ($n = $ 7 for each group) were fasted for 24 h before the experiment, then fixed in a 50 -ml centrifuge tube, and immersed in a $20 \pm 2$ °C water bath for 1 h up to the depth of the xiphoid process. After general anesthesia, saliva samples were taken in a germ-free centrifugal tube for 20 min. Saliva samples were 10,000 MW filtered and diluted before assay. The concentrations of nitrate in the saliva and serum were measured by quantification of total nitric oxide and nitrate/nitrite according to the manufacturer's protocol (R&D Systems)[34]. Nitrate concentration was determined by the optical density (O.D.) of each well using a microplate reader set at 540 nm (wavelength correction at 690 nm). The concentrations of total sialic acid were determined using a sialic acid assay kit (Abnova), in which sialic acid is oxidized to formylpyruvic acid, which reacts with thiobarbituric acid to form a pink-colored product. The color intensity at 549 nm is directly proportional to sialic acid concentration in the sample.

**Whole-genome sequencing**. Livers from six mice (sgRNA two-cell embryo #10, #17, #18 and wt #1, #2, #3) were harvested and digested completely for genomic DNA extraction (Animal Tissues/Cells Genomic DNA Extraction Kit, Solarbio). Genomic DNA was sheared to lengths between 200 and 700 bp with Covaris (LE220), and fragments ranging between 400 and 600 bp were selected for the libraries, which were prepared from 100 ng of DNA using the DNA LT Sample Prep Kit (Truseq; FC-121-4002). Libraries were sequenced on Illumina HiSeq X Ten instruments with a paired-end $1 \times 150$ -bp read length at an average of $30 \times$ coverage. Sequencing reads were aligned to the C57BL/6J GRCm38 (mm10) mouse reference genome using BWA-MEM (v0.7.12) as follows: bwa mem -t 32 -M. In order to improve the run time, aligned BAM files were split by chromosome, read pairs aligned to different chromosomes were filtered by SAMtools (version 1.5), and duplicates were marked using Picard tools (version 2.9.2). The alignments were

then processed using the Genome Analysis Toolkit (GATK-4.0.1.0) pipeline to produce a joint genotyped VCF file. SNPs and indels were extracted from the GATK joint genotyping file. Hard filtering was carried out on the joint samples according to the GATK recommendations with the following criteria: "QD < 2.0 || FS > 200.0 || ReadPosRankSum < −20.0 || SOR > 10.0." SNPs and indels were called with Mutect2 default settings. Sequencing and bioinformatics analysis were carried out at OE Biotech Ltd., Shanghai.

**Cytokine staining and cell sorting of Treg cells.** Mice were killed by cervical dislocation. A single-cell suspension was achieved by passing the spleen through a 70-μm nylon cell strainer (BD Biosciences). The single-cell suspension was washed in PBS and resuspended in erythrocyte lysis buffer (Qiagen) to remove the red blood cells. Anti-CD4 (GK1.5) (1:200), anti-CD3 (17A2) (1:200), anti-CD25 (PC61) (1:100), anti-CD127 (A7R34) (1:100), anti-CTLA4-PE (M290) (1:100), and anti-Foxp3 (MF-14) (1:100) antibodies were used to identify T-cell populations. For intracellular staining of Foxp3, the Foxp3-Staining Buffer Set (eBioscience) was used according to the manufacturer's protocol. CD3 + CD4 + CD25 + CD127- Treg cells were sorted with a FACSAriaII (BD Biosciences) and cultured in the 1640 medium (Corning) with 10% FBS, anti-CD3 (3 μg/mL, coated) and CD28 (2 μg/mL) antibody, and 20 ng/mL IL-2. The data were collected using a FAC-SAriaII (BD Biosciences), and analyzed with FlowJo software (Tree Star).

**Analysis of expression from RNA-seq.** Sequencing reads were mapped to the C57BL/6J GRCm38 (mm10) mouse reference genome using HISAT software. After multi-mapped reads were removed, FPKM values were calculated using Cuffnorm software. Sequencing and bioinformatics analysis were conducted by OE Biotech Ltd., Shanghai.

**Single-cell PCR.** Whole-genome amplification for single cell was carried out using the MALBAC® Single Cell WGA Kit (Yikon). The amplification product was used as a template for two rounds of PCR amplification using Ex Taq™ Version 2.0. Both rounds of the PCR program were: 95 °C for 5 min; 32 cycles of (95 °C for 30 s, 62 °C for 30 s, 72 °C for 50 s); 72 °C for 5 min; and 12 °C thereafter. In total, 0.5 μl of first-round PCR product was used as the template for the second-round PCR. The primers for genotyping are described in Supplementary Table 6.

**Reagents.** Reagents used in this study, related to experimental procedures, are described in Supplementary Table 7.

**Quantification of ISH, IHC, and statistical analysis.** Images were captured with a Marianas Yokogawa type spinning disk confocal microscope (Intelligent Imaging Innovations). Image analysis for ISH and immunohistochemistry was performed using FIJI software[35]. Image analysis for glomerulus diameter was performed using Image J software[36]. Values reported in all analyses were expressed as mean ± SEM. T test was used to determine the levels of difference between the groups and P-values for significance. T test was performed to compare the following: concentration of total sialic acid and $NO_3^-$, *Slc17a5* and *Virma* mRNA expression, glomerulus diameters, sialin, TMEM16A and AQP5 protein expression, frequencies of CD25(+)CD127(−) and Foxp3(+)Treg cells, and frequencies of CTLA-4(+) cells. All statistical analyses were carried out using the Graph Prism (v.7). Unless otherwise specified, all experiments were repeated at least three times.

**Reporting summary.** Further information on research design is available in the Nature Research Reporting Summary linked to this article.

## Data availability
The source data underlying Figs. 4a, b, 5a–e, 6a, c, d are provided as a Source Data file. Whole-genome sequencing data from this study are available through the NCBI Sequence Read Archive under accession no. PRJNA543729. The RNA high-throughput sequencing data reported in this paper has been deposited in the NCBI Sequence Read Archive (accession no. PRJNA543953). A reporting summary for this article is available as a Supplementary Information file. All other relevant data are available from the authors upon reasonable request.

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

## Acknowledgements

We are grateful to the Genetically Engineered Models team in the Laboratory Animal Center, Capital Medical University, for their technical support. We thank Dr. Haoyi Wang (Chinese Academy of Sciences) for his valuable comments on our paper. Dr. Debbie DeLoach provided editorial assistance. This study was supported by a grant from the National Natural Science Foundation of China (91649124) and grants from Beijing Municipality Government grants (Beijing Scholar Program PXM2018_014226_000021, PXM2017_014226_000023, PXM2018_193312_000006_0028S643_FCG PXM2016_014226_000034, PXM2016_014226_000034, PXM2016_014226_000006, PXM2015_014226_000116).

## Author contributions

S.W. and B.C. conceived and supervised the project. Y.W., J. Lu, B.P., and T.Z. performed experiments related to the production of knockout mice, with help from Y.L., X.F., J. Liu and J. Li, while Y.W., J. Lu, B.C. and J.Z. performed experiments related to whole-genome sequencing. J.Z., Y.W. and B.C. performed experiments related to in situ hybridization and immunohistochemistry. Y.W., D.T. and D.Z. performed Treg cell isolations and tests. The initial experiments were performed by Y.W. All authors contributed to data interpretation and data analysis. Y.W. and S.W. wrote the paper with input from J.Z. and B.C.

## Additional information

**Competing interests:** The authors declare no competing interests.

