## [Peer Review File · Nature Communications]

Reviewers' Comments:

Reviewer #1:

Remarks to the Author:

In this manuscript, Wu et al. described a one-step one-blastomere CRISPR/Cas9 injection method to rapidly generate viable chimeric founders carrying a heritable embryonically or postnatally lethal mutation. By analyzing the chimeric mutant founders, novel gene functions could be discovered. In addition, this method could also generate homozygous mutant progeny by mouse mating strategy, same as the strategy used by ES cells. This study confirmed a previous strategy published in Cell Research (doi:10.1038/cr.2017.58), in which, the authors reported the establishment and optimization of 2-cell embryo-CRISPR-Cas9 injection (2CC) method to study the in vivo function of essential genes in founder chimeric mice. Overall, this is an interesting study that may be suitable for publication in Nature Communications if the authors could address the following major questions.

1. The authors showed that direct zygote CRISPR/Cas9 injection targeting *Virma* could not generate mutant founders. Is this due to the inappropriate time point for injection? Whether injection of CRISPR/Cas9 targeting *Virma* at late-stage zygotes could generate *Virma* mutant chimeras? If this is possible, injecting at late-stage zygotes could likely lead to mosaic founders with high-percentage of chimerism, facilitating the subsequent phenotypic analysis and germline transmission.
2. Generating chimeric founders with simple or clear genotypes are particularly important for the phenotypic analysis both in founders and progeny. In this one-step one-blastomere CRISPR/Cas9 injection method, what's the percentage of founders carrying simple or clear genotypes? Does injection of Cas9 ribonucleoprotein complex have any effect on simplifying the the outcome of the genotype?
3. The efficiency of germline transmission of lethal mutation to F1 progeny shown in this manuscript is about 15%~35%. How many of these F1 progeny carry the same expected mutations existing in F0 founders? In addition, are there any measures that could be taken to improve the germline transmission efficiency in order to reduce the cost and time for screening F1 progeny?
4. Phenotype analyses in this manuscript were superficial. It's better to provide more evidence to demonstrate the gene functions.

Reviewer #2:

Remarks to the Author:

In this manuscript, Yi Wu et al., describe a method to generate chimeric mice bearing heritable embryonic and postnatal lethal mutations. To achieve this, the authors developed a single-step protocol that consists in the microinjection of the Cas9 mRNA (or protein) together with sgRNAs into one blastomere of a two-cell mouse embryo. These embryos are then reimplanted into pseudopregnant females to obtain F0 chimeric mice that are viable and can be used to study the role of the mutated gene in a postnatal context. The authors further show that these F0 chimeric mice can be used as founders to generate F1 homozygous or heterozygous knockout mutants to better study the role of embryonically mutant genes during development. To validate their method, the authors generated chimeric F0 mutants for different genes known to be embryonically or postnatally lethal when knocked-out. Among these, they characterized a new role for *VIRMA* protein in regulating kidney metabolism in adult mice and a role for *Sialin* protein in sialic acid and nitrate secretion of acinar cells.

Overall the manuscript is well written and the results do showcase the interest of this method in studying lethal mutations, an issue that has not been fully addressed in the literature. However, some important aspects of the approach need to be better characterized so that readers and potential users can evaluate whether it is suitable to their candidate genes. Furthermore, although the authors clearly present all the advantages of their method, they do not discuss its limitations, which I think is an important point to cover. For example, is this method suitable to all types of

lethal mutations, can the phenotypes observed at the F0 generation result from an interaction between wild-type and knockout cells within the same tissue and thus be biased?

Below you will find some major and minor points that should, in my opinion, be addressed by the authors.

Major points:

- In the introduction, the authors mention that the two-step microinjection method recently developed by Wang et al (Cell Research 2017) is only suitable for postnatally lethal mutations and does not allow the generation of F1 knockout mice with embryonically lethal mutations. Wang and colleagues did indeed only test a neonatally lethal mutation (Tet3 gene). However, since both methods rely on genome editing in one blastomer of a two-cell embryo and the generation of founder chimeric mice, it is not completely clear to me why the method described by Wang et al is not suitable for embryonically lethal mutations as well. Furthermore, Wang and collaborators were also able to show heritable mutations from founder mice to the F1 progeny. Could the authors clarify this point and clearly describe in their manuscript why the single-step microinjection method allows the generation of embryonically lethal mutants while the two-step method does not?

- Sanger sequencing of the targeted loci from chimeric mice presented in Figure 1 and 2 suggests that there is little to no sequence variability in the mutated alleles. However, the data is limited to the genotyping of the mice tails so it is possible that other types of mutations could be found in other tissues. This appears to be the case in Figure 3a where the Sanger sequencing of the Slc17a5 locus from mouse tail and liver shows two different mutation patterns. It would therefore be important to perform a PCR amplification of the targeted locus from different tissues (brain, liver, heart...) and test what is the variability in the sequence of the edited alleles among the different tissues. This is an important test because if users want to genotype the mice tails to screen for mice bearing frameshift mutations (in the case they want to study a specific phenotype at the F0 generation) they have to be sure that the mutation pattern found in the tail is the same throughout the animal.

In addition to this, it would be also interesting to measure the percentage of alleles that are edited in each tissue to know if the editing efficiency is homogeneous. This can be easily estimated from the Sanger sequencing electropherogram using TIDE analysis (<https://tide.deskgen.com> from Brinkman et al., 2014 Nucl. Acids Res).

- Similarly to the heterogeneity of editing between tissues, it is not clear whether cells carrying the mutation have both alleles mutated or if they can be heterozygous with one wt and one mutant allele. In situ hybridization and immunohistochemistry results presented in Figure 4 appear to show an overall decrease in VIRMA and Sialin proteins but it is not easy to tell whether there are cells that express those proteins at wt levels and others that do not express the proteins at all. Do the authors have any additional data to show if the single microinjection protocol leads to bi-allelic editing in targeted cells?

- As mentioned by authors in line 129-130 (Figure 1A and table 1), sgRNA targeting efficiency can significantly differ between two different sgRNAs. It might therefore be possible to use a relatively inefficient sgRNA to generate viable chimeric F0 mice carrying lethal mutations. To test this, the authors separately tested two different sgRNAs designed against the Virma gene that were co-injected with Cas9 mRNA into zygotes. Using this approach, the authors were not able to obtain newborn mice, although 160 zygotes were reimplanted. This result is interesting and surely validates the need for an alternative method to generate such F0 mice. However, the authors did not explain how the 2 different sgRNAs were selected. When using an in silico prediction tool with the two sequences provided by the authors, both of the sgRNAs appear to display an excellent efficiency score based on the "Doench" (Doench JG et al., 2016 Nature Biotechnology) and the "Moreno-Mateos" (Moreno-Mateos MA et al., 2015 Nature Methods) efficiency scores as well as the "out-of-frame" prediction score (Bae S et al., 2014 Nature Methods).

I therefore think that it would be useful to repeat the experiment using one or two sgRNAs with poor predicted efficiency scores and test whether under those settings the authors are still unable to obtain viable chimeric F0 mice upon Cas9/sgRNA zygote injection.

Below are the efficiency predictions for each sgRNA:

sgRNA#1 (used in Figure 1a): Doench score: 92 percentile; Moreno-Mateos score: 87% percentile; Bae score: 55

sgRNA#2 (used in Figure 1a): Doench score: 100 percentile; Moreno-Mateos score: 92 percentile; Bae score: 54

- The results obtained with Ctl4-4 (Figure 2E-G) are potentially interesting as conditional knockout models have viability issues. However, the data presented in Figure 2E-F is too superficial to conclude whether the one-step microinjection of two-cell embryo method is really an improvement over the conditional knock-out approach. The authors mention that 3 out of 9 mice carrying the mutation died around 21 days after birth (similarly to mice with a conditional Ctl4-4 knockout in CD4+ T Cells). What about the remaining 6 mice that survive past 21 days, do they have a specific phenotype similar to that described by Klocke K and collaborators upon conditional knockout of Ctl4-4 in adult mice by injection of tamoxifen (Klocke K et al., 2016 PNAS)? Since CTLA-4 is a cell surface marker, the authors should test whether the expression of CTLA-4 is indeed reduced in Foxp3+ Treg cells from the 6 mice that survived. Currently, it is not possible to know if the 6 mice that survived had a significant fraction of their T cells edited. If editing is not observed in T cells of those mice then this represents a limitation of the blastomer injection that should be mentioned.

Minor comments:

- Line 163: The authors wrote: "suggesting that Virma is embryonically lethal when knocked out in mice". However in line 130-131 the authors mentioned that "We chose Virma, a gene that is embryonically lethal when knocked out" and cited a paper to support this claim (Dickinson ME et al., 2016 Nature). The authors should therefore change the word "suggesting" in line 163 to "confirming" since they are confirming the findings from a previous work.

- Line 255-256: The authors mention that they only found 6 off-target sites upon whole-genome sequencing of 3 mutant mice. However, the sentence is misleading as the Venn-diagram shows that there probably are more than 6 off-target sites if we take all the individual SNPs that are common to 2 mice into account. The authors should therefore rephrase the sentence to mention that among all SNPs detected, 6 were in common between the three mice.

- Off-target search was performed by whole-genome sequencing of three mutant and three wild-type mice. The methods section mentions that the SAMtools mpileup function was used to identify SNPs and indels, however no details were given regarding the statistical analysis of the results generated by SAMtools. Knowing that not all the cells of the sampled tissue are edited and that the efficiency of indels at off-target sites is lower than the one at the targeted site, it would be interesting for readers to know the technical details and statistical analysis that allowed the identification of such loci as real indels and SNPs instead of sequencing errors.

- Line 102-104: The authors cite a paper (Jin et al., 2013 Free Radic Biol Med) regarding the use of low-doses of Cas9 mRNA/sgRNA but the actual paper that is referenced describes the role of salivary nitrate secretion against stress and does not mention anything related to the CRISPR-Cas9 technology. The authors should correct the bibliography to include the correct citation.

- Supplementary figure 1: The band without cleavage by EcoRV corresponds to the black arrow and not to the red arrow as stated in the figure legend. Furthermore, it would be preferable to put the two arrows on the right side of the same gel so that it is easier to understand.

Responds to referees:

Reviewer #1:

1. The authors showed that direct zygote CRISPR/Cas9 injection targeting *Virma* could not generate mutant founders. Is this due to the inappropriate time point for injection? Whether injection of CRISPR/Cas9 targeting *Virma* at late-stage zygotes could generate *Virma* mutant chimeras? If this is possible, injecting at late-stage zygotes could likely lead to mosaic founders with high-percentage of chimerism, facilitating the subsequent phenotypic analysis and germline transmission.

Response: Thanks for your suggestion. After 42 zygotes, post pronuclear fusion, were microinjected with Cas9 mRNA and sgRNA1 into the oviducts of 2 pseudopregnant females, no mice were born. This indicates that zygote microinjection using different zygote development stages is an ineffective method for generating viable F0 mice with embryonically lethal mutations. We added pertinent information to the revised manuscript.

2. Generating chimeric founders with simple or clear genotypes are particularly important for the phenotypic analysis both in founders

and progeny. In this one-step one-blastomere CRISPR/Cas9 injection method, what's the percentage of founders carrying simple or clear genotypes? Does injection of Cas9 ribonucleoprotein complex have any effect on simplifying the the outcome of the genotype?

Response: We are sorry that we did not provide enough information about the percentage of founders carrying clear frameshift mutations. The percentage of *Virma* or *Dpm1* founders carrying clear frameshift mutation is 100%. The percentage of *Slc17a5* founders carrying frameshift mutations is 57%, and *Ctla-4's* is 33%. Moreover, the injection of Cas9 ribonucleoprotein complex had no effect on the percentage of founders carrying clear frameshift mutations (*Slc17a5*, 27%; *Ctla-4*, 37.5%). We added pertinent information to Tables 1 and S4 in the revised manuscript.

3. The efficiency of germline transmission of lethal mutation to F1 progeny shown in this manuscript is about 15%~35%. How many of these F1 progeny carry the same expected mutations existing in F0 founders? In addition, are there any measures that could be taken to improve the germline transmission efficiency in order to reduce the cost and time for screening F1 progeny?

Response: Thank you for your suggestion. Yes, using TIDE analysis to measure the F0 founders' indel frequencies was beneficial. Those results showed that the founders with higher indel frequencies efficiently transmitted lethal mutations to F1 progeny. We added detailed information to Table S2, and results of germline transmission of *Ctla-4* founders in the revised version. The mutation patterns found in the tails of founders produced by microinjecting the CRISPR/Cas9 system into two-cell embryos is the same throughout the animal (Figure 3 in the revised manuscript). We found that 100% of F1 progeny carry the same, expected mutations found in the founders' tails.

4. Phenotype analyses in this manuscript were superficial. It's better to provide more evidence to demonstrate the gene functions.

Response: That is a good suggestion. Upon completion of additional experiments, we found novel phenotypes in adult founder mice generated by the one-step manipulation of two-cell embryos method. *Virma* knockout leads to focal segmental glomerulosclerosis, and RNA sequencing showed that notable gene clusters include cellular responses to vitamin and positive regulation of macrophage chemotaxis in *Virma* knockout animals.

Moreover, *Ctla-4* knockout leads to a preferential expansion of Treg cells in adult mouse spleens. Detailed information is in the revised manuscript.

Reviewer #2:

1. In the introduction, the authors mention that the two-step microinjection method recently developed by Wang et al (Cell Research 2017) is only suitable for postnatally lethal mutations and does not allow the generation of F1 knockout mice with embryonically lethal mutations. Wang and colleagues did indeed only test a neonatally lethal mutation (Tet3 gene). However, since both methods rely on genome editing in one blastomer of a two-cell embryo and the generation of founder chimeric mice, it is not completely clear to me why the method described by Wang et al is not suitable for embryonically lethal mutations as well. Furthermore, Wang and collaborators were also able to show heritable mutations from founder mice to the F1 progeny. Could the authors clarify this point and clearly describe in their manuscript why the single-step

microinjection method allows the generation of embryonically lethal mutants while the two-step method does not?

Response: Apologies for the inaccurate description. Wang et al. (2017) reported that founder chimeric mice of the postnatally lethal *Tet3* gene were successfully generated using the two-step microinjection of two-cell embryos method. The mutations from founder mice were heritable to the F1 progeny. However, embryonic lethality accounted for more than 80% of all mice with lethal genes post knockout. The team in that study did not test if their method succeeded in producing knockout mice with embryonically lethal mutations.

To increase the percentage (~50%) of KO cells in mice, the procedures of the two-step microinjection method first included the injection of Cas9 mRNA during the zygote stage and then a second injection of sgRNA and Cre mRNA into one blastomere during the two-cell embryo stage. However, a high percentage of lethally mutant cells in embryos may have caused the failure to obtain knockout F0 mice. In our study of the one-step microinjection method, the ability to produce viable founder mice with embryonically lethal mutations (*Virma* or *Dpm1*) was obviously less

than that of producing F0 mice with postnatally lethal knockouts (*Slc17a5* or *Ctla-4*). We added more details to the revised manuscript.

2. - Sanger sequencing of the targeted loci from chimeric mice presented in Figure 1 and 2 suggests that there is little to no sequence variability in the mutated alleles. However, the data is limited to the genotyping of the mice tails so it is possible that other types of mutations could be found in other tissues. This appears to be the case in Figure 3a where the Sanger sequencing of the *Slc17a5* locus from mouse tail and liver shows two different mutation patterns. It would therefore be important to perform a PCR amplification of the targeted locus from different tissues (brain, liver, heart...) and test what is the variability in the sequence of the edited alleles among the different tissues. This is an important test because if users want to genotype the mice tails to screen for mice bearing frameshift mutations (in the case they want to study a specific phenotype at the F0 generation) they have to be sure that the mutation pattern found in the tail is the same throughout the animal.

In addition to this, it would be also interesting to measure the percentage of alleles that are edited in each tissue to know if the editing efficiency is homogeneous. This can be easily estimated from the Sanger sequencing electropherogram using TIDE analysis (<https://tide.deskgen.com> from Brinkman et al., 2014 Nucl. Acids Res).

Response: Your suggestion really improved our manuscript. We performed a PCR amplification of the targeted loci (*Kiaa1429*, *Ctla-4*, and *Slc17a5*) from different tissues (tail, brain, liver, heart, lung, spleen, salivary glands, and Treg cells in spleen). The results showed that the mutation pattern found in the tails of founders produced by one-step microinjecting the CRISPR/Cas9 system into two-cell embryos is the same throughout the animal, and editing efficiency by TIDE analysis is homogeneous in each tissue.

3. Similarly to the heterogeneity of editing between tissues, it is not clear whether cells carrying the mutation have both alleles mutated or if they can be heterozygous with one wt and one mutant allele. In situ hybridization and immunohistochemistry results presented in Figure 4 appear to show an overall decrease in VIRMA and Sialin proteins but it is not easy to tell whether there are cells that express

those proteins at wt levels and others that do not express the proteins at all. Do the authors have any additional data to show if the single microinjection protocol leads to bi-allelic editing in targeted cells?

Response: Your suggestion really improved our manuscript. We isolated a single CD3(+)CD4(+)CD25(+)CD127(-) Treg cell from spleen tissue of one *Ctla-4* knockout F0 mouse generated by the OSTCM method and performed single-cell PCR, followed by sequencing analysis. The results showed that the single microinjection protocol led to the same bi-allelic editing in targeted cells as in tail tissue. Detailed information is in the revised manuscript.

4. As mentioned by authors in line 129-130 (Figure 1A and table 1), sgRNA targeting efficiency can significantly differ between two different sgRNAs. It might therefore be possible to use a relatively inefficient sgRNA to generate viable chimeric F0 mice carrying lethal mutations. To test this, the authors separately tested two different sgRNAs designed against the *Virma* gene that were co-injected with Cas9 mRNA into zygotes. Using this approach, the authors were not able to obtain newborn mice, although 160 zygotes were reimplanted. This result is interesting and surely validates the need for an alternative method to generate such F0

mice. However, the authors did not explain how the 2 different sgRNAs were selected. When using an in silico prediction tool with the two sequences provided by the authors, both of the sgRNAs appear to display an excellent efficiency score based on the “Doench” (Doench JG et al., 2016 Nature Biotechnology) and the “Moreno-Mateos”

(Moreno-Mateos MA et al., 2015 Nature Methods) efficiency scores as well as the “out-of-frame” prediction score (Bae S et al., 2014 Nature Methods).

I therefore think that it would be useful to repeat the experiment using one or two sgRNAs with poor predicted efficiency scores and test whether under those settings the authors are still unable to obtain viable chimeric F0 mice upon Cas9/sgRNA zygote injection.

Below are the efficiency predictions for each sgRNA:

sgRNA#1 (used in Figure 1a): Doench score: 92 percentile;

Moreno-Mateos score: 87% percentile; Bae score: 55

sgRNA#2 (used in Figure 1a): Doench score: 100 percentile;

Moreno-Mateos score: 92 percentile; Bae score:54

Response: That is a good idea. We selected another sgRNA (sgRNA3) whose score of on-target efficiency is 0.450 based on

the CRISPRko prediction tool (Doench JG et al., 2016 Nature Biotechnology). Although we transplanted more than 98 early two-pronuclear zygotes microinjected with Cas9 mRNA and sgRNA3 into the oviducts of 4 pseudopregnant females, no mice were born. sgRNA selection was dependent on on-target efficiency and off-target potentials. Using the CRISPRko prediction tool, the sgRNAs' on-target efficiency scores differed (sgRNA1, 0.771; sgRNA2, 0.645; and sgRNA3, 0.450). Their Off-Target ranks differed (sgRNA1, 185; sgRNA2, 11; and sgRNA3, 107), and smaller numbers represented lower potentials of sgRNA off-target. Their Combined Ranks with on-target efficiency and off-target effect obviously differed (sgRNA1, 167; sgRNA2, 4; and sgRNA3, 232). Therefore, our results indicated that zygote microinjection using different sgRNA sequence features was an ineffective method for generating viable F0 mice with embryonically lethal mutations. Detailed information is in the revised manuscript.

5. The results obtained with Ctla-4 (Figure 2E-G) are potentially interesting as conditional knockout models have viability issues. However, the data presented in Figure 2E-F is too superficial to conclude whether the one-step microinjection of two-cell embryo method is really an improvement over the conditional knock-out

approach. The authors mention that 3 out of 9 mice carrying the mutation died around 21 days after birth (similarly to mice with a conditional *Ctla-4* knockout in CD4⁺ T Cells). What about the remaining 6 mice that survive past 21 days, do they have a specific phenotype similar to that described by Klocke K and collaborators upon conditional knockout of *Ctla-4* in adult mice by injection of tamoxifen (Klocke K et al., 2016 PNAS)? Since CTLA-4 is a cell surface marker, the authors should test whether the expression of CTLA-4 is indeed reduced in Foxp3⁺ Treg cells from the 6 mice that survived. Currently, it is not possible to know if the 6 mice that survived had a significant fraction of their T cells edited. If editing is not observed in T cells of those mice then this represents a limitation of the blastomer injection that should be mentioned.

Response: That is a good observation. Like the adult mouse model of inducible *Ctla-4* deletion reported by Klocke K et al. (PNAS, 2016), the frequencies of CD25(+)CD127(-) or Foxp3(+) Treg cells were significantly elevated in the spleens of adult F0 knockout mice generated using the OSTCM method. That method led to a reduction of CTLA-4 levels in CD3(+)CD4(+)CD25(+)CD127(-) Treg cells, regardless of whether the in vitro treatment (aCD3: 3ug/mL, aCD28: 2ug/mL, and IL2: 20

ng/mL for 3 days) was used. Detailed information is in the revised manuscript.

Minor comments:

1. Line 163: The authors wrote: “suggesting that Virma is embryonically lethal when knocked out in mice”. However, in line 130-131 the authors mentioned that “We chose Virma, a gene that is embryonically lethal when knocked out” and cited a paper to support this claim (Dickinson ME et al., 2016 Nature). The authors should therefore change the word “suggesting” in line 163 to “confirming” since they are confirming the findings from a previous work.

Response: Thanks for your suggestion. “Suggesting” was replaced by “confirming” in the revised manuscripts.

2. Line 255-256: The authors mention that they only found 6 off-target sites upon whole-genome sequencing of 3 mutant mice. However, the sentence is misleading as the Venn-diagram shows that there probably are more than 6 off-target sites if we take all the individual SNPs that are common to 2 mice into account. The

authors should therefore rephrase the sentence to mention that among all SNPs detected, 6 were in common between the three mice.

Response: Thanks for your suggestion. We rephrased this misleading sentence in the revised manuscript, replacing it with: “With this analysis, we found six common off-targets among three mutant mice”.

3. Off-target search was performed by whole-genome sequencing of three mutant and three wild-type mice. The methods section mentions that the SAMtools mpileup function was used to identify SNPs and indels, however no details were given regarding the statistical analysis of the results generated by SAMtools. Knowing that not all the cells of the sampled tissue are edited and that the efficiency of indels at off-target sites is lower than the one at the targeted site, it would be interesting for readers to know the technical details and statistical analysis that allowed the identification of such loci as real indels and SNPs instead of sequencing errors.

Response: Thanks for your suggestion. Detailed information is in the revised manuscript.

4. Line 102-104: The authors cite a paper (Jin et al., 2013 Free Radic Biol Med) regarding the use of low-doses of Cas9 mRNA/sgRNA but the actual paper that is referenced describes the role of salivary nitrate secretion against stress and does not mention anything related to the CRISPR-Cas9 technology. The authors should correct the bibliography to include the correct citation.

Response: Thanks for your suggestion. We have corrected the mistakenly cited paper (Jin et al., 2013 Free Radic Biol Med). It is now (Wang et al., 2013 Cell) in the revised manuscript.

5. Supplementary figure 1: The band without cleavage by EcoRV corresponds to the black arrow and not to the red arrow as stated in the figure legend. Furthermore, it would be preferable to put the two arrows on the right side of the same gel so that it is easier to understand.

Response: Thanks for your suggestion. The band without cleavage by EcoRV corresponds to the black arrow, and the band with cleavage by EcoRV corresponds to the red arrow. The two arrows were placed on the right side of the same gel in the revised manuscript.

6. Furthermore, although the authors clearly present all the advantages of their method, they do not discuss its limitations, which I think is an important point to cover. For example, is this method suitable to all types of lethal mutations, can the phenotypes observed at the F0 generation result from an interaction between wild-type and knockout cells within the same tissue and thus be biased?

Response: Thanks for your suggestion. We have included detailed information on this topic in the discussion of the revised manuscript.

Reviewers' Comments:

Reviewer #1:

Remarks to the Author:

The authors have addressed all my concerns appropriately and I believe that the revised manuscript can be considered for publication in Nature Communications now.

Reviewer #2:

Remarks to the Author:

The authors have satisfactorily answered most of my requests. I think that the manuscript has been significantly improved and that it is now suitable for publication.

I do have, nevertheless, some minor comments on the new version of the manuscript:

- Line 186: It is not clear that you are referring to the classical zygote microinjection method and not to the OSTCM approach. Could the authors rewrite the sentence so this is clear?

- Line 206 - 208: This sentence is redundant with the previous one. If the authors want to conclude that their method is suitable to transmit the editing to the next generation for most postnatally lethal mutations then they should put the sentence at the end of line 216.

- Line 236 - 245: I thank the authors for testing the overall editing in different tissues but I think that this information should not be mentioned in this paragraph (it does not really correspond to the what it is announced in the subtitle of the section. Maybe the authors could place it at the end of line 216 when they describe the transmission of the edits to the F1 generation?

- Line 259: The reference to the CRISPOR tool should be put after the word "CRISPOR" on line 258.

- Line 288 – 292: The results of the RNA-seq are very interesting. However, the authors failed to mention whether Virma is among the differentially expressed genes. Also, the figure is too small and the resolution is not sufficient for readers to zoom in and look at the differentially expressed genes.

- Finally, the RNA-seq data does not seem to have been deposited in a public repository yet.

Responses to referees:

Reviewer #2:

Minor comments:

1. Line 186: It is not clear that you are referring to the classical zygote microinjection method and not to the OSTCM approach. Could the authors rewrite the sentence so this is clear?

Response: Thanks for your suggestion. We rephrased this sentence in the revised manuscript.

2. Line 206 - 208: This sentence is redundant with the previous one. If the authors want to conclude that their method is suitable to transmit the editing to the next generation for most postnatally lethal mutations then they should put the sentence at the end of line 216.

Response: Thank you for your suggestion – it really improved our manuscript.

3. Line 236 - 245: I thank the authors for testing the overall editing in different tissues but I think that this information should not be

mentioned in this paragraph (it does not really correspond to the what it is announced in the subtitle of the section. Maybe the authors could place it at the end of line 216 when they describe the transmission of the edits to the F1 generation?

Response: Thank you – we followed your suggestion.

4. Line 259: The reference to the CRISPOR tool should be put after the word “CRISPOR” on line 258.

Response: Thank you – we made this correction in the manuscript.

5. Line 288 – 292: The results of the RNA-seq are very interesting. However, the authors failed to mention whether *Virma* is among the differentially expressed genes. Also, the figure is too small and the resolution is not sufficient for readers to zoom in and look at the differentially expressed genes.

Response: Thank you for your comments. We added a high-resolution version of the figure to the revised manuscript and a table of the differentially expressed genes to the Supplementary Information section. RNA-seq was unable to detect differences in *Virma* mRNA expression and coverage of mapped reads between wt mice and knockout F0 mice, which suggested that the *Virma* transcripts containing premature termination codons generated by

frameshift mutations were not degraded efficiently (Reber S, *et al.*, Mol Biol Cell, 2018 29, 75-83). However, q-PCR using specific primer targeted the *Virma* mutant region demonstrated that *Virma* mRNA expression in kidney tissue of F0 mice generated by our OSTCM method decreased by about 50% compared with wt mice. Moreover, individual clones were sequenced from RT-PCR products and the results showed that frameshift mutations were present in about 50% of *Virma* mRNA in the kidney tissue of knockout F0 mice. These results suggest that the *Virma* gene was knocked out in about 50% of cells in the kidney tissue of knockout F0 mice generated by our OSTCM method.

6.the RNA-seq data does not seem to have been deposited in a public repository yet.

Response: We have deposited the RNA-seq data and provided detailed access information to the manuscript.